# PRISM: Partial-label Relational Inference with Spatial and Spectral Cues

**Yiyang Gu**[1*], **Wenrui Wu**[2*], **Yifang Qin**[1], **Taian Guo**[1], **Tao Zhe**[3], **Jiaru Tang**[4], **Zhiping Xiao**[5†],
**Weizhi Zhang**[6], **Ziyue Qiao**[7], **Wei Ju**[1], **Dongjie Wang**[3], **Xiao Luo**[8†], **Philip S. Yu**[6], **Ming Zhang**[1†]

[1] State Key Laboratory for Multimedia Information Processing,
School of Computer Science, PKU-Anker LLM Lab, Peking University
[2] School of Mathematical Sciences, Peking University
[3] Department of Electrical Engineering and Computer Science, University of Kansas
[4] School of Psychological and Cognitive Sciences, Peking University
[5] Paul G. Allen School of Computer Science and Engineering, University of Washington
[6] Department of Computer Science, University of Illinois Chicago
[7] School of Computing and Information Technology, Great Bay University
[8] Department of Statistics, University of Wisconsin–Madison
`yiyanggu@pku.edu.cn, wenruiwu@stu.pku.edu.cn, patxiao@uw.edu,`
`xiaoluo@wisc.edu, mzhang_cs@pku.edu.cn`

## Abstract

In many real-world scenarios, acquiring precise labels for graph-structured data is expensive or even infeasible, as reliable annotation often requires substantial expert knowledge or computational resources. As a result, graph labels are often noisy and ambiguous. This challenge motivates partial-label graph learning, where each graph is weakly annotated with a candidate label set containing the true label. However, such ambiguous supervision makes it hard to extract reliable graph semantics and increases the risk of overfitting to noisy candidate labels. To address these challenges, we propose a unified framework named PRISM that performs relational inference with spatial and spectral cues to alleviate the impact of label ambiguity. On the one hand, PRISM captures discriminative spatial cues by aligning prototype-guided substructures across graphs. On the other hand, it decomposes graph signals into multiple frequency bands and extracts global spectral cues with an attention mechanism, which preserve frequency-specific semantics. We integrate these complementary views into a hybrid relational graph and perform an iterative label propagation under candidate constraints. Extensive experiments on a range of well-known datasets demonstrate that PRISM consistently outperforms strong baselines under various noise settings.

## 1 Introduction

Graph-structured data is pervasive across diverse domains such as drug discovery, molecular property prediction, social network analysis, and recommendation systems (Fang et al., 2022; Zhang et al., 2021b; Wang et al., 2021). To handle such complex structures, Graph Neural Networks (GNNs) (Welling & Kipf, 2016; Hamilton et al., 2017; Xu et al., 2018; Zhang et al., 2021a; Yang et al., 2025) have emerged as powerful tools for learning expressive graph representations. GNNs typically operate by recursively aggregating information from node neighborhoods and summarizing node embeddings via global pooling (Gao & Ji, 2019; Lee et al., 2021). They have achieved state-of-the-art results in a wide range of applications, including biomedical classification (Wu et al., 2022; Li et al., 2022a), cross-modal retrieval (Chen et al., 2022), and event understanding (Liu et al., 2023).

Despite these advances, existing graph classification frameworks are still highly data hungry: they require accurate labeled samples to learn a discriminative classifier (Li et al., 2022b; Rousseau et al.,

---

*Equal contribution.
†Corresponding authors.

2015). However, in many practical scenarios, acquiring ground-truth labels is expensive or technically infeasible. For instance, identifying the properties of chemical compounds often depends on density functional theory (DFT) simulations (Becke, 2014), which can be expensive and time-consuming. To alleviate this problem, some automated tools are often employed to assist with the annotation process. However, this approach can introduce label ambiguity, as different tools may lead to inconsistent annotations. These noisy labels can pose significant challenges to the training of graph classifiers. Although self-supervised graph learning methods such as GraphCL (You et al., 2020) have alleviated the reliance on labels by using contrastive objectives to learn graph representations during pre-training, they still depend on accurate labels to train a classifier during downstream classification stages. Their performance tends to degrade significantly under label ambiguity.

In this paper, we focus on a practical yet underexplored setting: Partial-label Graph Learning (PLGL). Each graph is annotated with a candidate label set that contains the true class label, but the exact ground-truth is unknown. This situation commonly arises when labels are generated from automated tools or coarse taxonomy (Ge et al., 2022; Gu et al., 2024). While partial label learning has been explored in computer vision (Feng & An, 2019; Lyu et al., 2020), PLGL presents unique challenges due to the structural complexity and non-Euclidean nature of graphs. First, ambiguous supervision introduces semantic uncertainty, making it difficult to capture class-discriminative substructures. Second, GNNs tend to overfit noisy signals, particularly when candidate sets contain semantically similar labels. Third, graphs exhibit patterns at multiple structural resolutions, from local motifs to global topology, which cannot be effectively captured by global pooling. Recent works have explored weakly supervised graph classification using pseudo-labeling (Ju et al., 2023b; Gu et al., 2025b) or contrastive learning (Yue et al., 2022; Luo et al., 2023), but they face two limitations: (i) reliance on overconfident predictions, leading to error accumulation under label ambiguity; and (ii) lack of explicit use of structural and spectral diversity to distinguish candidate labels.

To bridge this gap, we introduce a unified framework PRISM for Partial-label Relational Inference with Spatial and Spectral Modeling. PRISM alleviates the impact of label ambiguity through three mechanisms. First, it aligns prototype-guided substructures across graphs to extract *spatial cues*, which can uncover class-discriminative local patterns against the label ambiguity. Second, it decomposes graph signals into multiple frequency bands to encode *spectral cues*. We propose a multi-band attention mechanism that preserves frequency-specific semantics critical for global structural understanding. Third, a *hybrid relational graph* is constructed to integrate both spatial and spectral cues. We then employ a label propagation mechanism to refine supervision signals iteratively. We further utilize a momentum-based update of a soft label confidence matrix under candidate constraints, which suppresses noise accumulation and facilitates robust optimization.

The contributions of this paper can be summarized in three points: (1) **Understudied Problem.** We study an understudied problem of partial-label graph learning, which has extensive practical applications in various domains. (2) **Novel Framework.** We propose a novel relational inference framework named PRISM. It leverages relational propagation to integrate spatial and spectral cues for label disambiguation. (3) **Extensive experiments.** We validate PRISM across diverse benchmarks and demonstrate its superiority over existing weakly supervised and graph learning approaches.

## 2 BACKGROUND

**Problem Definition.** Let $\mathcal{G} = \{G_i = (\mathcal{V}_i, \mathcal{E}_i, \boldsymbol{X}_i)\}_{i=1}^N$ denote a collection of $N$ graphs, where each graph $G_i$ consists of a node set $\mathcal{V}_i$, edge set $\mathcal{E}_i$, and node features $\boldsymbol{X}_i \in \mathbb{R}^{|\mathcal{V}_i| \times d}$. We denote by $\boldsymbol{A}_i \in \{0,1\}^{|\mathcal{V}_i| \times |\mathcal{V}_i|}$ the adjacency matrix of $\mathcal{E}_i$. For each graph $G_i$, we are given a candidate label set $\mathcal{S}_i \subset \mathcal{Y}$, where $\mathcal{Y} = \{1, 2, \ldots, C\}$ is the complete label space. The candidate set $\mathcal{S}_i$ includes the true label $y_i^*$ but does not reveal which one is correct. Our objective is to learn a graph classifier $f(G_i; \theta)$ that predicts the ground-truth label $y_i^*$ for each graph in the test set, by training only on ambiguous candidate sets without access to ground-truth supervision.

**Graph Neural Networks.** Graph Neural Networks (GNNs) are widely used to encode graph structures by recursively aggregating information from node neighborhoods. At each layer $l$, the representation of node $v$ is updated by combining its own embedding with messages from its neighbors:

$$\boldsymbol{h}_v^{(l)} = \phi^{(l)} \left( \boldsymbol{h}_v^{(l-1)}, \psi^{(l)} \left( \{\boldsymbol{h}_u^{(l-1)}\}_{u \in \mathcal{N}(v)} \right) \right), \tag{1}$$

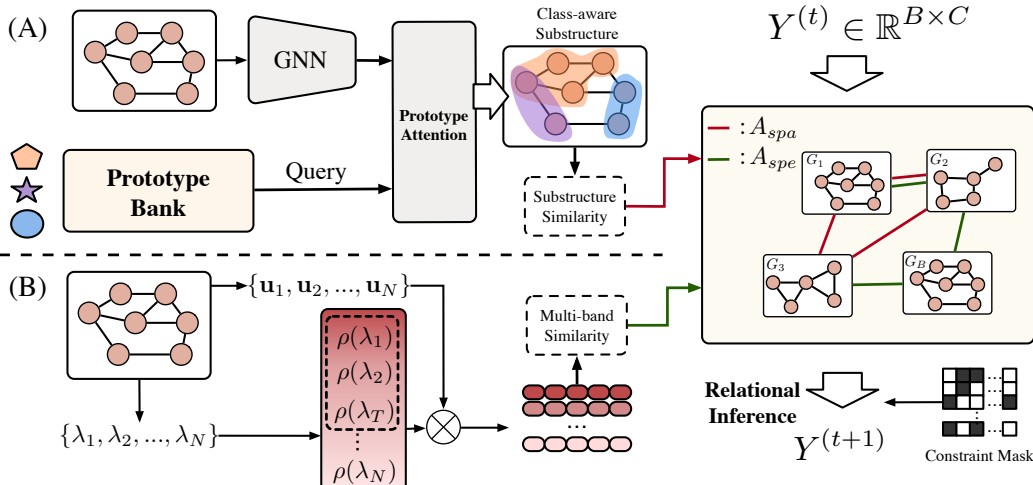

Figure 1: Overview of PRISM. Our framework jointly models spatial substructures (A) and spectral frequency patterns (B) to disambiguate partial labels. Prototype-guided attention and multi-band spectral encoding form a relational graph for iterative label refinement under candidate constraints.

where $\psi^{(l)}$ and $\phi^{(l)}$ are the aggregation function and the combination function, respectively, and $\mathcal{N}(v)$ denotes the neighbor set of node $v$. After $L$ layers, node embeddings $\{\boldsymbol{h}_v^{(L)}\}_{v \in \mathcal{V}}$ are aggregated into a graph-level representation using a readout function:

$$\boldsymbol{g} = \text{READOUT}\left(\{\boldsymbol{h}_v^{(L)}\}_{v \in \mathcal{V}}\right),$$ (2)

where READOUT can be sum, mean, or pooling methods. These graph representations provide the basis for downstream tasks such as graph-level classification.

## 3 METHODOLOGY

This paper introduces a novel framework PRISM for partial-label graph learning that integrates local substructural cues and global spectral cues to resolve candidate label ambiguities. Each graph is encoded through two complementary pathways: a spatial encoder captures discriminative spatial cues by aligning prototype-guided substructures across graphs, while a spectral encoder decomposes graph signals into distinct frequency bands, capturing both smooth and non-smooth structural patterns. These dual pathways induce a relational graph with two types of edges: one encoding prototype-based substructure similarity, the other reflecting spectral affinity across graphs. We then perform an iterative propagation over this graph to fuse information from both relational types, which enables the model to distill consistent label cues from noisy candidates and refine the supervision signals. The overview of PRISM is illustrated in Figure 1.

### 3.1 SPATIAL CUES VIA SUBSTRUCTURE MATCHING

In partial-label graph learning, the true label of each instance is hidden within a noisy candidate set, where multiple labels may be semantically correlated. Relying solely on global graph representations may blur these distinctions, since graphs with overlapping candidate labels can exhibit a similar overall topology. In contrast, local substructures frequently encode the most discriminative evidence for class separation. Motivated by this observation, we design a structure-aware disambiguation module that aligns prototype-guided substructures across related instances, which helps uncover consistent label semantics under ambiguity.

To incorporate class-level semantic priors, we maintain a momentum-updated *prototype bank* $\{\boldsymbol{p}_c \in \mathbb{R}^F\}_{c \in \mathcal{Y}}$, where each prototype $\boldsymbol{p}_c$ tracks the aggregated global representation of graphs associated with candidate label $c$. At training step $j$, we update each prototype as:

$$\boldsymbol{p}_c^{(j)} \leftarrow m \cdot \boldsymbol{p}_c^{(j-1)} + (1 - m) \cdot \frac{1}{|\mathcal{B}_c|} \sum_{i \in \mathcal{B}_c} \boldsymbol{g}_i,$$ (3)

where $\boldsymbol{g}_i$ denotes the global embedding of graph $G_i$, $m \in (0, 1)$ is the momentum coefficient, and $\mathcal{B}_c$ is the set of current-batch graphs containing label $c$ in their candidate set. Based on the prototype bank, we apply a *prototype-guided attention mechanism* to extract $C$ substructure embeddings per graph, where each embedding is aligned with a candidate class. Given the node embeddings from the final GNN layer $\{\boldsymbol{h}_v^{(L)}\}_{v \in \mathcal{V}_i}$ and the full prototype set $\{\boldsymbol{p}_c\}_{c=1}^C$, attention weights are computed to obtain class-aware latent components:

$$\boldsymbol{r}_i^{(c)} = \sum_{v \in \mathcal{V}_i} \alpha_{vc} \cdot \boldsymbol{h}_v^{(L)}, \text{ where } \alpha_{vc} = \frac{\exp(\boldsymbol{h}_v^{(L)\top} \boldsymbol{p}_c)}{\sum_{v'} \exp(\boldsymbol{h}_{v'}^{(L)\top} \boldsymbol{p}_c)}. \tag{4}$$

The resulting embeddings $\{\boldsymbol{r}_i^{(c)}\}_{c=1}^C$ serve as interpretable and class-specific substructures for downstream comparison. We then construct a relational graph over graph pairs that share at least one candidate label. Let $\mathcal{P} = \{(G_i, G_j) \mid \mathcal{S}_i \cap \mathcal{S}_j \neq \emptyset, \ i \neq j\}$ denote the set of such graph pairs. For each pair $(G_i, G_j) \in \mathcal{P}$, we define a prototype-aware substructure similarity:

$$s_{ij}^{spa} = \max_{c \in \mathcal{S}_i \cap \mathcal{S}_j} \cos\left(\boldsymbol{r}_i^{(c)}, \boldsymbol{r}_j^{(c)}\right) \cdot \cos\left(\frac{\boldsymbol{r}_i^{(c)} + \boldsymbol{r}_j^{(c)}}{2}, \boldsymbol{p}_c\right). \tag{5}$$

For each graph, we retain its top-$k_a$ neighbors with the highest $s_{ij}^{spa}$ scores to form a sparse relational graph with a normalized adjacency matrix $\boldsymbol{A}^{spa}$ that encodes substructure-level agreement under label semantics. This spatial reasoning module provides a fine-grained structural information, thereby enhancing label disambiguation by promoting relational consistency among structurally aligned graph instances.

## 3.2 SPECTRAL CUES VIA MULTI-BAND FREQUENCY ATTENTION

While the spatial disambiguation module focuses on extracting local spatial cues, graph spectra offer a complementary global perspective by capturing both low-frequency smoothness and high-frequency irregularities. However, many existing spectral methods combine spectral components through linear aggregation across eigenmodes (Yang et al., 2022; Bo et al., 2023). Such aggregation mixes signals from different frequencies into a single representation, which may blur structurally diverse signals and obscure frequency-specific patterns that are crucial for graph understanding. We thus propose a *Multi-Band Frequency Attention* module, which not only synthesizes spectral information into a unified form but also retains frequency-specific characteristics by explicitly modeling bands. Specifically, our method: (1) preserves the resolution of each frequency band by constructing explicit band-wise representations, and (2) provides fine-grained cross-graph reasoning from band-level similarity comparison. This architecture allows for frequency-aware graph embeddings, which concentrate on informative spectral patterns while suppressing the noise of less crucial bands.

Given a graph $G$ with normalized Laplacian eigenvalues $\{\lambda_1, \lambda_2, \ldots, \lambda_N\}$ and corresponding eigenvectors $\{\boldsymbol{u}_1, \boldsymbol{u}_2, \ldots, \boldsymbol{u}_N\}$, we first project scalar spectral values into a learnable signal space using harmonic expansion:

$$\rho(\lambda) = [\sin(k\lambda), \cos(k\lambda)]_{k=1}^K \cdot \boldsymbol{W}_\rho, \tag{6}$$

where $\boldsymbol{W}_\rho \in \mathbb{R}^{2K \times d}$ is a shared learnable projection matrix, and $K$ is the number of periods. For efficiency, we only consider the $T$ smallest eigenvectors, therefore producing $T$ distinct frequency embeddings, where each embedding serves as a filtered spectral descriptor of the graph. To construct band-specific node representations, we modulate the $p$-th eigenvector $\boldsymbol{u}_p \in \mathbb{R}^N$ with its associated harmonic encoding $\rho(\lambda_p)$ to form:

$$\boldsymbol{X}^{(p)} = \boldsymbol{u}_p \otimes \rho(\lambda_p) \in \mathbb{R}^{N \times d}, \tag{7}$$

where $\otimes$ denotes the outer product broadcast across all nodes. We then apply a multi-layer neural network $f_{\text{shared}}$, shared across all frequency bands, to each $\boldsymbol{X}^{(p)}$ to extract spectral signals. For each band $p \in \{1, \ldots, T\}$, we compute the corresponding graph-level embedding as:

$$\boldsymbol{z}^{(p)} = \text{READOUT}\left(f_{\text{shared}}(\boldsymbol{X}^{(p)})\right), \tag{8}$$

where READOUT denotes a readout function that integrates node embeddings. This yields a set of band-level embeddings $\{\boldsymbol{z}^{(1)}, \ldots, \boldsymbol{z}^{(T)}\}$ that capture spectrally filtered representations of the graph

across multiple frequency perspectives. To synthesize multi-scale spectral signals, we employ a soft attention mechanism across bands:

$$\boldsymbol{z} = \sum_{p=1}^{T} \alpha^{(p)} \boldsymbol{z}^{(p)}, \text{where } \alpha^{(p)} = \frac{\exp\left(\boldsymbol{a}^{\top}\sigma(\boldsymbol{W}\boldsymbol{z}^{(p)})\right)}{\sum_{q=1}^{T} \exp\left(\boldsymbol{a}^{\top}\sigma(\boldsymbol{W}\boldsymbol{z}^{(q)})\right)}. \tag{9}$$

Here, $\boldsymbol{W}$ and $\boldsymbol{a}$ are learnable parameters, and $\sigma(\cdot)$ denotes a nonlinear activation function. For cross-graph spectral reasoning, we compute band-wise similarity between graphs. Let $G_i$ and $G_j$ denote two graphs with band embeddings $\{\boldsymbol{z}_i^{(p)}\}_{p=1}^{T}$ and $\{\boldsymbol{z}_j^{(p)}\}_{p=1}^{T}$, respectively. Their similarity is defined as:

$$s_{ij}^{spe} = \max_{p \in \{1,...,T\}} \cos\left(\boldsymbol{z}_i^{(p)}, \boldsymbol{z}_j^{(p)}\right). \tag{10}$$

To ensure label-aware alignment, we restrict edges to graph pairs with overlapping candidate sets, i.e., $\mathcal{S}_i \cap \mathcal{S}_j \neq \emptyset$. Each graph links to its top-$k_e$ most similar neighbors, forming a relational graph with a normalized adjacency matrix $\boldsymbol{A}^{spe}$ that supports label disambiguation from the spectral perspective.

We theoretically establish that both $\boldsymbol{A}^{spa}$ and $\boldsymbol{A}^{spe}$ contribute to label disambiguation. Intuitively, the distribution of hidden node embeddings should be determined by the prototype of the true label to a certain degree. Building on this observation, we present the following theorem with proof provided in Appendix A:

**Theorem 1.** *Assume* $\mathbb{E}\left[h_v^{(i)}|y_i^* = c\right] = p_c$, $\forall v \in \mathcal{V}_i$, $\eta_{jk}^{(i)} = \mathbb{P}\left(\boldsymbol{A}_i\left(j,k\right) = 1\right)$ *is random variable whose distribution can be determined by* $y_i^*$, *and* $\min_{1 \leq i \leq N, 1 \leq p \leq T}\left|\lambda_{p+1}^{(i)} - \lambda_p^{(i)}\right| \geq \delta$ *for some* $\delta > 0$. *Then, we have for any* $i, j \in \{1, 2, ..., N\}$:

$$\mathbb{P}\left(A_{ij}^{spa} = 1 | y_i^* = y_j^*\right) \to 1 \tag{11}$$

*and*

$$\mathbb{P}\left(A_{ij}^{spe} = 1 | y_i^* = y_j^*\right) \to 1 \tag{12}$$

*as* $|\mathcal{V}_i|, |\mathcal{V}_j| \to \infty$.

Theorem 1 suggests that if graph $i$ and $j$ shares the same true label, the probabilities of $A_{ij}^{spa}$ and $A_{ij}^{spe}$ being 1 will both converge to 1, which works on the subsequent label propagation with disambiguation from noisy candidate sets.

## 3.3 LABEL DISAMBIGUATION VIA RELATIONAL INFERENCE

Based on the relational graph constructed in previous modules, which encodes spatial proximity and spectral correlation as distinct relational types, we develop an iterative label propagation framework to enhance supervision under partially labeled settings where ground-truth labels are inaccessible. This framework integrates complementary structural and spectral signals to refine soft supervision progressively, while rigorously enforcing candidate label constraints throughout the entire refinement process to ensure both semantic validity and training stability.

Specifically, we denote the initial label matrix by $\boldsymbol{Y}^{(0)} \in \mathbb{R}^{N \times C}$. At each iteration $e$, the label matrix is updated using two normalized adjacency matrices, i.e., $\boldsymbol{A}^{spa}$ and $\boldsymbol{A}^{spe}$. The propagation rule is:

$$\tilde{\boldsymbol{Y}}^{(e+1)} = \mu \cdot \boldsymbol{Y}^{(e)} + (1 - \mu) \cdot \mathcal{N}\left(\boldsymbol{A}^{spa}\boldsymbol{Y}^{(e)} + \boldsymbol{A}^{spe}\boldsymbol{Y}^{(e)}\right), \tag{13}$$

where $\mu \in (0, 1)$ controls the update momentum, and $\mathcal{N}(\cdot)$ denotes row-wise $\ell_1$ normalization. We further apply a binary mask $\boldsymbol{M} \in \{0, 1\}^{N \times C}$ to constrain label assignments to valid candidate classes, where $\boldsymbol{M}_{ic} = 1$ if and only if $c \in S_i$ and $\boldsymbol{M}_{ic} = 0$ otherwise:

$$\boldsymbol{Y}^{(e+1)} = \mathcal{N}\left(\tilde{\boldsymbol{Y}}^{(e+1)} \odot \boldsymbol{M}\right), \tag{14}$$

where $\odot$ denotes element-wise multiplication. After $E$ iterations, the refined label matrix $\boldsymbol{Y}^{(E)}$ captures multi-relational consistency while remaining faithful to partial supervision, enabling effective disambiguation of noisy candidate labels.

We further maintain a soft label confidence matrix $\boldsymbol{Q} \in \mathbb{R}^{N \times C}$ to stabilize optimization, which is uniformly initialized on the candidate label set. We utilize the soft labels $\boldsymbol{Y}^{(E)}$ inferred from relational propagation to update $\boldsymbol{Q}$ gradually. The update follows an exponential moving average (EMA) scheme:

$$\boldsymbol{Q}_i \leftarrow \mathcal{N}\left(\beta \cdot \boldsymbol{Q}_i + (1-\beta) \cdot \boldsymbol{Y}_i^{(E)}\right), \tag{15}$$

where $\beta \in (0,1)$ is the momentum coefficient. This mechanism promotes stable refinement of the supervision signals during training.

## 3.4 UNIFIED TRAINING OBJECTIVE

We adopt a unified objective that couples the soft confidence target with the candidate constraint to train our framework effectively. Specifically, we first apply an MLP-based classifier to the spatial-view embeddings $\boldsymbol{g}$ to obtain the final predictions $\boldsymbol{P}^{spa} \in \mathbb{R}^{N \times C}$. In parallel, we employ a distinct classifier that projects the spectral-view embeddings $\boldsymbol{z}$ to $\boldsymbol{P}^{spe} \in \mathbb{R}^{N \times C}$. The training loss for the spatial (spectral) view is defined as the negative marginal log-likelihood over the candidate label set:

$$\mathcal{L}_{sup}^{(o)} = -\frac{1}{B}\sum_{i=1}^{B}\log\sum_{c \in \mathcal{S}_i}\boldsymbol{P}_{ic}^{(o)}\boldsymbol{Q}_{ic}, \quad o \in \{spa, spe\}, \tag{16}$$

where $\mathcal{S}_i$ is the candidate label set for sample $i$, and $B$ is the batch size. This formulation encourages the model to align predictions with the soft label confidence within the candidate label set. We then optimize the spatial and spectral objectives jointly to capture complementary graph characteristics:

$$\mathcal{L} = \mathcal{L}_{sup}^{spa} + \mathcal{L}_{sup}^{spe}. \tag{17}$$

This dual-view supervision facilitates robust label disambiguation. We further offer a theoretical analysis of the proposed method, particularly focusing on the convergence of the label confidence matrix and training loss under certain conditions. To begin with, let $Y^* = (Y_i^*)_{i=1}^{N} \in \{0,1\}^{N \times C}$ be the matrix consisting of ground-truth one-hot label vector. Denote the classifier as $f_{classifier}$ which produces the label prediction matrix $P$, the final predicted label confidence vector of graph $G_i$ is $P_i = f_{classifier}(\boldsymbol{g}_i)$. If the classifier is well-trained, it should recover label $c$ from $\boldsymbol{p}_c$ since $\boldsymbol{p}_c$ is the prototype of label $c$. Based on this, we have the following results with the proof in Appendix B:

**Theorem 2.** *Under the assumption of Theorem 1, further assume $f_{classifier}(\boldsymbol{p}_c) = \mathbb{I}_c, \forall c \in \mathcal{Y}$ where $\mathbb{I}_c \in \{0,1\}^{C}$ denotes one-hot vector whose $c$-th component is 1 while the rest are 0, we have:*

$$Q_i \xrightarrow{a.s.} Y_i^* \tag{18}$$

*as $|\mathcal{V}_i|, E \to \infty$. And*

$$\mathbb{E}[\mathcal{L}_{sup}] \to 0 \tag{19}$$

*as $\min_{1 \leq i \leq N}|\mathcal{V}_i|, E \to \infty$.*

Theorem 2 indicates that if each graph has enough node information and we conduct enough iterations, the soft label confidence matrix updated by EMA will converge to the ground-truth and the training loss will tend to zero, which further implies our framework can resolve label ambiguity by aligning prototype-guided substructures across graphs. The condition $|\mathcal{V}_i| \to \infty$ can be replaced by $B, N \to \infty$ to a certain degree, since graphs with the same label and candidate label set can be regarded as a whole, and the whole number of nodes tends to infinity when $B, N \to \infty$.

## 3.5 COMPUTATIONAL EFFICIENCY ANALYSIS

Let $|\mathcal{V}|$ be the number of nodes, $|\mathcal{E}|$ the number of edges, $d$ the feature dimension, $L$ the number of GNN layers, and $T$ the number of spectral bands. In preprocessing, we compute the $T$ smallest eigenvectors and their spectral encodings, which are reused throughout training. During training, the spatial view performs message passing with complexity $\mathcal{O}(L|\mathcal{E}|d)$, while the spectral view operates on pre-computed features across $T$ bands and applies a shared MLP, resulting in a total cost of $\mathcal{O}(T|\mathcal{V}|d)$. Since $L$ and $T$ are small constants, the overall training complexity is $\mathcal{O}(|\mathcal{E}|d)$, which is linear in the number of edges and consistent with standard GNN-based methods.

Table 1: The compared accuracy (mean%±std%) on different graph classification datasets. The best results are highlighted in boldface and the second best results are underlined. $q = P(\overline{y} \in Y | \overline{y} \neq y)$ reflecting the level of label ambiguity.

| Datasets | ENZYMES | | Letter-High | | COIL-DEL | | CIFAR10 | | COLORS-3 | |
|---|---|---|---|---|---|---|---|---|---|---|
| Methods | $q = 0.3$ | $q = 0.5$ | $q = 0.3$ | $q = 0.5$ | $q = 0.05$ | $q = 0.1$ | $q = 0.3$ | $q = 0.5$ | $q = 0.3$ | $q = 0.5$ |
| GCN | $48.44_{\pm2.06}$ | $40.22_{\pm2.93}$ | $44.00_{\pm1.08}$ | $35.94_{\pm1.82}$ | $50.43_{\pm1.07}$ | $41.63_{\pm1.74}$ | $43.68_{\pm0.68}$ | $41.35_{\pm0.65}$ | $74.87_{\pm0.25}$ | $60.67_{\pm1.64}$ |
| GAT | $49.11_{\pm2.93}$ | $34.67_{\pm3.87}$ | $61.33_{\pm3.48}$ | $53.04_{\pm3.06}$ | $59.77_{\pm1.97}$ | $46.63_{\pm1.54}$ | $52.93_{\pm1.22}$ | $48.54_{\pm0.46}$ | $71.83_{\pm0.22}$ | $62.56_{\pm3.54}$ |
| GIN | $47.11_{\pm4.59}$ | $34.22_{\pm1.78}$ | $50.43_{\pm1.92}$ | $35.59_{\pm3.75}$ | $46.23_{\pm0.88}$ | $37.29_{\pm1.04}$ | $43.91_{\pm0.45}$ | $41.24_{\pm0.52}$ | $48.17_{\pm0.44}$ | $41.00_{\pm2.75}$ |
| GraphSAGE | $47.33_{\pm3.03}$ | $39.33_{\pm3.11}$ | $70.96_{\pm1.48}$ | $60.35_{\pm1.83}$ | $58.91_{\pm1.92}$ | $49.23_{\pm1.90}$ | $51.92_{\pm0.26}$ | $47.44_{\pm0.83}$ | $71.24_{\pm3.00}$ | $56.63_{\pm5.81}$ |
| TopKPool | $44.22_{\pm2.76}$ | $36.00_{\pm4.80}$ | $55.25_{\pm2.74}$ | $43.83_{\pm5.21}$ | $44.83_{\pm2.19}$ | $34.63_{\pm2.08}$ | $48.97_{\pm1.24}$ | $42.87_{\pm1.31}$ | $56.69_{\pm3.58}$ | $33.49_{\pm1.74}$ |
| SAGPool | $46.67_{\pm2.53}$ | $37.11_{\pm5.00}$ | $55.71_{\pm4.71}$ | $39.30_{\pm5.49}$ | $41.89_{\pm4.28}$ | $30.17_{\pm1.85}$ | $50.01_{\pm0.68}$ | $45.16_{\pm0.36}$ | $59.91_{\pm4.14}$ | $24.62_{\pm0.32}$ |
| EdgePool | $51.11_{\pm3.06}$ | $33.33_{\pm1.99}$ | $64.17_{\pm2.44}$ | $55.36_{\pm2.16}$ | $56.74_{\pm3.98}$ | $45.89_{\pm1.30}$ | $50.17_{\pm0.64}$ | $45.90_{\pm0.44}$ | $76.96_{\pm0.13}$ | $62.31_{\pm1.23}$ |
| ASAP | $44.44_{\pm3.06}$ | $31.56_{\pm3.34}$ | $65.04_{\pm1.22}$ | $52.75_{\pm4.41}$ | $46.20_{\pm4.08}$ | $34.94_{\pm3.02}$ | $50.10_{\pm0.63}$ | $44.81_{\pm1.57}$ | $70.11_{\pm0.54}$ | $62.47_{\pm0.98}$ |
| Graph Transplant | $51.78_{\pm2.39}$ | $43.78_{\pm3.41}$ | $74.84_{\pm1.44}$ | $66.78_{\pm1.86}$ | $66.57_{\pm1.60}$ | $57.11_{\pm1.03}$ | $53.79_{\pm1.11}$ | $48.95_{\pm1.47}$ | $74.66_{\pm1.30}$ | $62.72_{\pm2.37}$ |
| PiCO | $46.88_{\pm2.76}$ | $35.78_{\pm3.02}$ | $73.56_{\pm1.71}$ | $64.63_{\pm4.35}$ | $76.25_{\pm1.66}$ | $63.69_{\pm1.42}$ | $53.47_{\pm1.14}$ | $46.04_{\pm1.20}$ | $53.99_{\pm0.92}$ | $34.74_{\pm1.77}$ |
| TGNN | $53.33_{\pm3.51}$ | $42.22_{\pm4.39}$ | $70.43_{\pm0.97}$ | $59.83_{\pm1.32}$ | $62.28_{\pm1.05}$ | $50.20_{\pm1.17}$ | OOM | OOM | $75.84_{\pm1.81}$ | $63.95_{\pm2.18}$ |
| GraphCL | $54.22_{\pm5.14}$ | $39.78_{\pm5.09}$ | $72.00_{\pm2.01}$ | $62.49_{\pm2.00}$ | $69.94_{\pm2.31}$ | $60.17_{\pm2.96}$ | $53.57_{\pm0.87}$ | $48.10_{\pm0.61}$ | $72.89_{\pm1.24}$ | $61.55_{\pm1.01}$ |
| GraphACL | $54.44_{\pm2.33}$ | $44.89_{\pm4.75}$ | $69.80_{\pm1.01}$ | $57.68_{\pm2.85}$ | $71.40_{\pm0.95}$ | $60.29_{\pm3.04}$ | $53.25_{\pm0.62}$ | $47.86_{\pm0.65}$ | $73.84_{\pm2.12}$ | $63.57_{\pm1.39}$ |
| DEER | $\underline{58.22}_{\pm2.18}$ | $\underline{47.56}_{\pm2.57}$ | $\underline{80.12}_{\pm1.26}$ | $\underline{72.29}_{\pm1.54}$ | $\underline{79.94}_{\pm1.20}$ | $\underline{68.03}_{\pm1.11}$ | $\underline{57.08}_{\pm0.57}$ | $\underline{52.48}_{\pm0.72}$ | $\underline{88.13}_{\pm2.08}$ | $\underline{66.81}_{\pm2.82}$ |
| **PRISM (Ours)** | $\mathbf{63.11}_{\pm0.83}$ | $\mathbf{51.33}_{\pm3.61}$ | $\mathbf{82.55}_{\pm0.89}$ | $\mathbf{78.32}_{\pm1.37}$ | $\mathbf{85.69}_{\pm1.06}$ | $\mathbf{79.48}_{\pm1.45}$ | $\mathbf{58.29}_{\pm0.83}$ | $\mathbf{55.10}_{\pm0.98}$ | $\mathbf{91.93}_{\pm2.26}$ | $\mathbf{80.57}_{\pm2.69}$ |

## 4 EXPERIMENTS

### 4.1 EXPERIMENTAL SETUP

**Datasets.** To comprehensively assess the performance of PRISM, we conduct experiments on five established graph classification benchmarks spanning the bioinformatics and vision domains: EN-ZYMES (Schomburg et al., 2004), Letter-High (Riesen & Bunke, 2008), COIL-DEL (Riesen & Bunke, 2008), CIFAR10 (Dwivedi et al., 2020), and COLORS-3 (Knyazev et al., 2019). Following (Gu et al., 2024), we randomly inject false positive labels into the candidate set to generate partial-label data. Specifically, each incorrect label $\overline{y} \neq y^*$ is included with probability $q = P(\overline{y} \in \mathcal{S} \mid \overline{y} \neq y^*)$, controlling the level of label ambiguity. A higher $q$ indicates noisier candidate supervision. We vary $q \in \{0.1, 0.3, 0.5\}$ for the majority of datasets, and $q \in \{0.02, 0.05, 0.1\}$ for COIL-DEL to account for its larger label space.

**Baselines.** We compare the proposed PRISM with a comprehensive set of baselines across multiple paradigms: (a) *Graph neural networks*: GCN (Welling & Kipf, 2016), GAT (Veličković et al., 2017), GIN (Xu et al., 2018), and GraphSAGE (Hamilton et al., 2017); (b) *Hierarchical graph pooling methods*: TopKPool (Gao & Ji, 2019), SAGPool (Lee et al., 2019), EdgePool (Diehl, 2019), and ASAP (Ranjan et al., 2020), all using GraphSAGE as the backbone; (c) *Graph augmentation method*: Graph Transplant (Park et al., 2022); (d) *Unsupervised contrastive graph learning*: GraphCL (You et al., 2020) and GraphACL (Luo et al., 2023); (e) *Semi-supervised graph learning*: TGNN (Ju et al., 2023b); (f) *Partial label learning in vision*: PiCO (Wang et al., 2022), adapted to the graph setting with a GraphSAGE encoder for fair comparison; (g) *Partial label learning for graphs*: DEER (Gu et al., 2024). Further details on the baselines are provided in the Appendix E.

**Implementation and Evaluation Protocol.** All experiments are implemented using PyTorch with the PyG backend. We use a two-layer GraphSAGE with 512 hidden units as the shared encoder across all methods. Training is performed using the Adam optimizer (Kingma & Ba, 2014) with an initial learning rate of 0.001 and a batch size of 128. All these results are averaged over five independent runs with individual seeds, each reporting mean accuracy and standard deviation. More details about the implementation are provided in the Appendix F.

### 4.2 PERFORMANCE COMPARISON

We report the quantitative results of our PRISM against competitive baselines under various ambiguity levels in Table 1. According to the results, we make the following observations: (1) Our PRISM consistently achieves superior performance across all datasets and various ambiguity settings. It surpasses all baselines by a considerable margin. For example, on ENZYMES, our method

Table 2: Ablation study on ENZYMES, Letter-High, and CIFAR10.

| Datasets | ENZYMES | | Letter-High | | CIFAR10 | |
|---|---|---|---|---|---|---|
| Variants | $q = 0.3$ | $q = 0.5$ | $q = 0.3$ | $q = 0.5$ | $q = 0.3$ | $q = 0.5$ |
| PRISM w/o Sub | $61.78_{\pm 2.06}$ | $49.56_{\pm 3.55}$ | $80.70_{\pm 1.20}$ | $76.11_{\pm 2.19}$ | $56.65_{\pm 1.01}$ | $53.30_{\pm 0.77}$ |
| PRISM w/o Spa | $60.89_{\pm 2.26}$ | $48.00_{\pm 3.87}$ | $79.65_{\pm 1.21}$ | $74.72_{\pm 0.99}$ | $55.23_{\pm 1.19}$ | $51.72_{\pm 1.08}$ |
| PRISM w/o Spe | $61.55_{\pm 1.94}$ | $48.89_{\pm 2.11}$ | $80.17_{\pm 0.43}$ | $76.46_{\pm 2.45}$ | $55.72_{\pm 0.93}$ | $52.65_{\pm 1.12}$ |
| PRISM w/o Rel. Infer | $57.78_{\pm 2.81}$ | $45.11_{\pm 3.95}$ | $78.43_{\pm 1.03}$ | $71.24_{\pm 2.18}$ | $53.61_{\pm 1.32}$ | $49.79_{\pm 1.27}$ |
| **PRISM (Full Model)** | $\mathbf{63.11}_{\pm 0.83}$ | $\mathbf{51.33}_{\pm 3.61}$ | $\mathbf{82.55}_{\pm 0.89}$ | $\mathbf{78.32}_{\pm 1.37}$ | $\mathbf{58.29}_{\pm 0.83}$ | $\mathbf{55.10}_{\pm 0.98}$ |

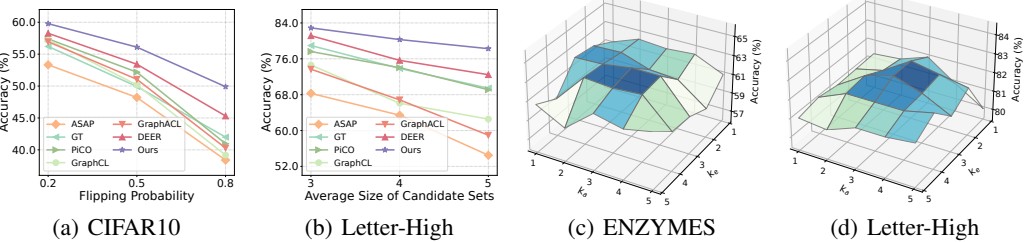

| (a) CIFAR10 | (b) Letter-High | (c) ENZYMES | (d) Letter-High |
|---|---|---|---|

Figure 2: (a) Performance comparison in scenarios with hierarchical label noise. (b) Performance comparison in scenarios with competitive label noise. (c)(d) Performance *w.r.t.* top-$k_a$ and top-$k_e$ on ENZYMES and Letter-High.

outperforms the second-best by $8.3\%$ under $q = 0.3$, confirming its robustness and generalization under weak supervision. (2) On fine-grained datasets such as COIL-DEL, where class granularity and semantic ambiguity are pronounced, PRISM retains a strong advantage over baseline methods. On COIL-DEL with $q = 0.1$, it attains $79.48\%$ accuracy, exceeding the prior best (DEER, $68.03\%$) by $16.8\%$, showing effective candidate label disambiguation via complementary spatial and spectral cues. (3) Compared to the previous partial label learning methods, our framework consistently improves performance across all graph datasets. Although PiCO with GraphSAGE encoders is adapted to graphs, it still performs poorly, demonstrating the limits of directly transferring CV-based methods and the necessity of graph-specific modeling. (4) As label ambiguity $q$ increases, most baselines degrade sharply, whereas our PRISM exhibits a much slower decline. This robustness stems from integrating spectral and substructural reasoning, with an iterative label propagation mitigating noisy and misleading supervision. More results under $q = 0.1$ are provided in Appendix D.

**Semantically Correlated Label Noise.** In many real-world scenarios, labels are not independent but semantically correlated, leading to candidate sets that contain noisy labels with stronger affinity to the ground-truth than unrelated classes, thereby introducing additional challenges for robust learning. An important question is whether PRISM can reliably address such semantically entangled label ambiguity. To evaluate this, we design two experimental protocols: (1) *Hierarchical label noise.* We exploit the coarse-to-fine taxonomy of CIFAR10 (i.e., vehicles vs. animals) and introduce noise by flipping negative labels within the same super-class as the true label with probability $q$, thereby forming semantically plausible candidates. (2) *Competitive label noise.* Following (Yan & Guo, 2023), we pretrain a graph neural network (GNN) on clean data to capture inter-class semantic dependencies. We then randomly select noisy candidates from the Top-$K$ predictions of this GNN, with $K = 6$ for Letter-High, and vary the candidate set size by adjusting sampling ratios. The results are presented in Figure 2 (a)(b). We can see that our PRISM consistently outperforms all baselines under both noise schemes across different ambiguity levels, thereby highlighting its robustness to semantically correlated label noise.

## 4.3 ABLATION STUDY

To assess the contribution of each core component in PRISM, we conduct ablation experiments on ENZYMES, Letter-High, and CIFAR10 under noise levels $q = 0.3$ and $q = 0.5$, with results in Table 2. Removing substructure alignment (PRISM w/o Sub) and replacing class-specific matching with holistic graph embeddings causes clear performance drops, especially under high ambiguity, showing that discriminative substructure cues outperform coarse comparisons. Excluding the spatial branch (PRISM w/o Spa) further reduces accuracy, as the absence of neighborhood-aware signals

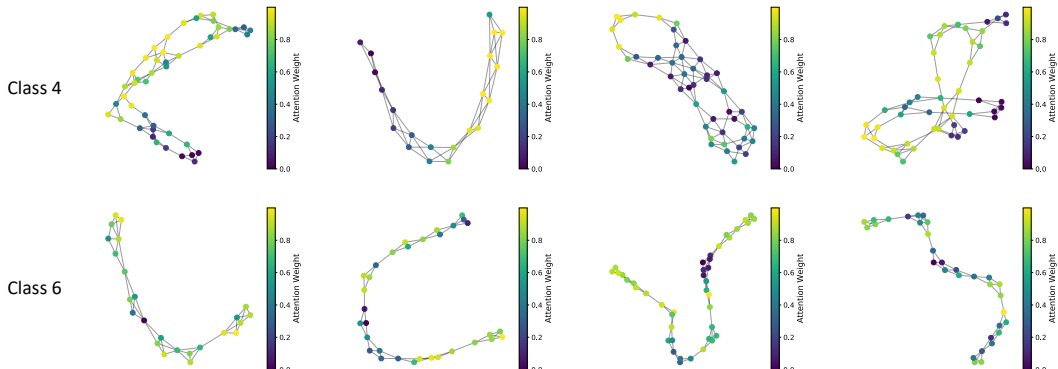

Figure 3: Class-specific attention maps reveal consistent focus on discriminative substructures across graph instances within different categories.

weakens the capture of fine-grained topology critical for resolving label ambiguity, particularly in CIFAR10. Discarding the spectral attention module (PRISM w/o Spe) also adversely impacts performance, since multi-band decomposition can capture global spectral characteristics, which complement local structure. Furthermore, removing the relational inference module (PRISM w/o Rel. Infer) yields the sharpest degradation, indicating that graph-of-graph modeling and label propagation are indispensable for alleviating the impact of label noise.

## 4.4 SENSITIVITY ANALYSIS

We analyze the effect of the number of neighbors retained in the spatial and spectral relational graphs, i.e., $k_a$ and $k_e$. The results are shown in Figure 2 (c)(d). We can see that accuracy generally increases as $k_a$ and $k_e$ grow from 1 to moderate values, then saturates or slightly declines. Maybe the reason is that adding structurally or spectrally aligned instances improves cross-graph consistency by enhancing the density of reliable signals. However, overly dense connectivity introduces noisy or weakly correlated neighbors, potentially causing label propagation drift. Our model achieves stable performance under a broad range of $k$ values, reflecting its robustness to hyperparameter choices in relational graph construction.

## 4.5 VISUALIZATION OF CLASS-SPECIFIC ATTENTION

We visualize class-specific attention maps from the ENZYMES dataset in Figure 3. Each subgraph represents a different graph instance, and the color of each node indicates its attention weight relative to the corresponding class prototype. Warmer colors signify a higher contribution to the class-specific substructure reasoning. Although the graph structures vary, nodes with high attention consistently correspond to meaningful substructures such as motifs, central connectors, or functional cores, while irrelevant or peripheral nodes receive lower attention. This consistent focus within each class highlights the model's ability to identify semantically aligned substructures across diverse graphs and yield human-interpretable views of functionally significant topological regions.

## 5 CONCLUSION

In this paper, we study the problem of partial-label graph learning, where every graph is associated with a noisy candidate label set. We propose a relational inference framework named PRISM to address the challenges of semantic uncertainty and structural complexity in this problem. PRISM first integrates spatial substructure alignment and spectral frequency modeling via a relational graph. We then leverage an iterative label propagation mechanism with candidate constraints to disambiguate supervision signals. Extensive experiments on a range of widely-used datasets validate the effectiveness and robustness of our PRISM under various ambiguity levels, highlighting its potential as a principled solution for learning from weakly supervised graph data.

ACKNOWLEDGEMENT

Ming Zhang and Yiyang Gu are supported by grants from the National Key Research and Development Program of China with Grant No. 2023YFC3341203 and the National Natural Science Foundation of China (NSFC Grant Number 62276002).

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

## A  PROOF OF THEOREM 1

To begin with, we denote adjacent matrix of graph $i$ as $A^{(i)}$ for the sake of convenience, take READOUT function as the mean in (2) that $g_i = \frac{1}{|\mathcal{V}_i|} \sum_{v \in \mathcal{V}_i} h_v^{(i)}$. Since $p_c$ is the prototype of label $c$, we assume the latent representations of nodes satisfy

$$\mathbb{E}\left[h_v^{(i)}|y_i^* = c\right] = p_c, \ \forall v \in \mathcal{V}_i. \tag{20}$$

where $y_i^*$ is the ground-truth label of graph $i$. Assume the latent representations of nodes in one graph are identically distributed, it follows from (20) and the law of large number that for any graph $i$ with true label $y_i^* = c$

$$g_i = \frac{1}{|\mathcal{V}_i|} \sum_{v \in \mathcal{V}_i} h_v^{(i)} \xrightarrow{a.s} \mathbb{E}\left[h_v^{(i)}|y_i^* = c\right] = p_c$$

as $|\mathcal{V}_i| \to \infty$, which is account for (3) to a certain degree. As for the adjacent matrix, we refer to (Gu et al., 2024) to consider the generalized random dot product graph, which can date back to (Solanki et al., 2021) that for graph $i$:

$$\mathbb{P}\left(A_{jk}^{(i)} = 1|\Theta\right) = \alpha_n \xi_j^T I_{r,-r} \xi_k$$

where $\Theta = (\xi_1, ..., \xi_n) \sim F^{(i)}, I_{r,-r} = \text{diag}\left(I_r, -I_r\right)$. We assume the distribution $F^{(i)}$ is determined by the global graph feature $g_i$ that there exist a continuous matrix function $\mathcal{M}$ such that

$$A^{(i)} = \mathcal{M}\left(g_i\right), \ \forall i \in \{1, 2, ..., N\}. \tag{21}$$

Take coefficients $\alpha_{vc}^{(i)} = \frac{1}{|\mathcal{V}_i|}$ and denote $\lambda_p^{(i)}, u_p^{(i)}$ as the $p$-th eigenvalue, eigenvector for graph $i$ for the sake of simplicity. First we consider if $y_i^* = c, y_j^* = c'$ then by (20) and the continuity theorem

$$\cos\left(r_i^{(c)}, r_j^{(c')}\right) = \sum_{v \in \mathcal{V}_i, v' \in \mathcal{V}_j} \alpha_{vc}^{(i)} \alpha_{v'c'}^{(j)} \cos\left(h_v^{(i)}, h_{v'}^{(j)}\right)$$

$$= \cos\left(\sum_{v \in \mathcal{V}_i} \frac{1}{|\mathcal{V}_i|} h_v^{(i)}, \frac{1}{|\mathcal{V}_j|} \sum_{v' \in \mathcal{V}_j} h_{v'}^{(j)}\right) \xrightarrow{a.s.} \cos\left(p_c, p_{c'}\right)$$

as $|\mathcal{V}_i|, |\mathcal{V}_j| \to \infty$, and

$$\cos\left(r_i^{(c)}, p_c\right) = \sum_{v \in \mathcal{V}_i} \alpha_{vc}^{(i)} \cos\left(h_v^{(i)}, p_c\right) = \cos\left(\frac{1}{|\mathcal{V}_i|} \sum_{v \in \mathcal{V}_i} h_v^{(i)}, p_c\right) \xrightarrow{a.s.} \cos\left(p_c, p_c\right) = 1,$$

as $|\mathcal{V}_i| \to \infty$, which further implies

$$\mathbb{P}\left(A_{ij}^{spa} = 1|y_i^* = y_j^*\right) = \sum_{c=1}^{C} \mathbb{P}\left(A_{ij}^{spa} = 1|y_i^* = y_j^* = c\right) \mathbb{P}\left(y_i^* = y_j^* = c|y_i^* = y_j^*\right)$$

$$\geq \sum_{c=1}^{C} \mathbb{P}\left(s_{ij}^{spa} = 1|y_i^* = y_j^* = c\right) \mathbb{P}\left(y_i^* = y_j^* = c|y_i^* = y_j^*\right)$$

$$\geq \sum_{c=1}^{C} \mathbb{P}\left(\cos\left(r_i^{(c)}, r_j^{(c)}\right) = \cos\left(\frac{r_i^{(c)} + r_j^{(c)}}{2}, p_c\right) = 1|y_i^* = y_j^* = c\right) \mathbb{P}\left(y_i^* = y_j^* = c|y_i^* = y_j^*\right)$$

$$\to \sum_{c=1}^{C} \mathbb{P}\left(y_i^* = y_j^* = c|y_i^* = y_j^*\right) = 1,$$

as $|\mathcal{V}_i|, |\mathcal{V}_j| \to \infty$, where the second and third line follow from the definitions of $A^{spa}$ and $s_{ij}^{spa}$. As for (12), observe that (21) indicates the adjacent matrix of graph $i$ given $y_i^* = c$ satisfies

$$A^{(i)} = \mathcal{M}\left(\frac{1}{|\mathcal{V}_i|} \sum_{v \in \mathcal{V}_i} h_v^{(i)}\right) \xrightarrow{a.s.} \mathcal{M}\left(p_c\right)$$

as $|\mathcal{V}_i| \to \infty$, which implies $||A^{(i)} - A^{(j)}||_F \xrightarrow{a.s.} 0$ given $y_i^* = y_j^* = c$. Further by Weyl's Perturbation Theorem in (Oudghiri, 2005) we have

$$\left| \lambda_p^{(i)} - \lambda_p^{(j)} \right| \leq ||A^{(i)} - A^{(j)}||_F \xrightarrow{a.s.} 0,$$

while by a variant of Davis-Kahan Theorem, Theorem 2 in (Yu et al., 2015) and $\min_{1 \leq i \leq N, 1 \leq p \leq T} \left| \lambda_{p+1}^{(i)} - \lambda_p^{(i)} \right| \geq \delta$ we have

$$||u_p^{(i)} - u_p^{(j)}|| \leq \frac{4}{\delta} ||A^{(i)} - A^{(j)}||_F \xrightarrow{a.s.} 0$$

for $1 \leq p \leq T$. Therefore by (8) and the continuity theorem we have

$$||z_i^{(p)} - z_j^{(p)}|| \xrightarrow{a.s.} 0$$

for $1 \leq p \leq T$, which leads to

$$1 \geq \cos\left( z_i^{(p)}, z_j^{(p)} \right) = \frac{||z_i^{(p)}||^2 + ||z_j^{(p)}||^2 - ||z_i^{(p)} - z_j^{(p)}||^2}{2||z_i^{(p)}|| \cdot ||z_j^{(p)}||} \xrightarrow{a.s.} \frac{||z_i^{(p)}||^2 + ||z_j^{(p)}||^2}{2||z_i^{(p)}|| \cdot ||z_j^{(p)}||} \geq 1.$$

Then by the definitions of $A^{spe}$ and $s_{ij}^{spe}$ we obtain

$$\mathbb{P}\left( A_{ij}^{spe} = 1 | y_i^* = y_j^* \right) = \sum_{c=1}^{C} \mathbb{P}\left( A_{ij}^{spe} = 1 | y_i^* = y_j^* = c \right) \mathbb{P}\left( y_i^* = y_j^* = c | y_i^* = y_j^* \right)$$

$$\geq \sum_{c=1}^{C} \mathbb{P}\left( s_{ij}^{spe} = 1 | y_i^* = y_j^* = c \right) \mathbb{P}\left( y_i^* = y_j^* = c | y_i^* = y_j^* \right)$$

$$\geq \sum_{c=1}^{C} \mathbb{P}\left( \cos\left( z_i^{(p)}, z_j^{(p)} \right) = 1, p = 1, ..., T | y_i^* = y_j^* = c \right) \mathbb{P}\left( y_i^* = y_j^* = c | y_i^* = y_j^* \right)$$

$$\to \sum_{c=1}^{C} \mathbb{P}\left( y_i^* = y_j^* = c | y_i^* = y_j^* \right) = 1,$$

as $|\mathcal{V}_i|, |\mathcal{V}_j| \to \infty$, which completes the proof. $\qquad\square$

## B    PROOF OF THEOREM 2

Recall that we assume the classifier is well-trained such that

$$f_{classifier}(\boldsymbol{p}_c) = \mathbb{I}_c, \ \forall c \in \mathcal{Y}, \tag{22}$$

where $\mathbb{I}_c \in \{0, 1\}^C$ denotes one-hot vector whose $c$-th component is 1 and the rest are 0. By Theorem 1 and (13) we have

$$Y_i^{(E)} \xrightarrow{a.s.} Y_i^*$$

as $|\mathcal{V}_i|, E \to \infty$, which further implies $Q_i \xrightarrow{a.s.} Y_i^*$ according to (15). By the continuity theorem and (22) we have

$$P_i = f_{classifier}(\boldsymbol{g}_i) = f_{classifier}\left( \sum_{v \in \mathcal{V}_i} \frac{1}{|\mathcal{V}_i|} h_v^{(i)} \right) \to f_{classifier}(\boldsymbol{p}_{y_i^*}) = \mathbb{I}_{y_i^*} = Y_i^* \tag{23}$$

as $|\mathcal{V}_i| \to \infty$. Denote $|\mathcal{V}| = \min_{1 \leq i \leq N} |\mathcal{V}_i|$, by dominated convergence theorem and (16) we obtain

$$\lim_{|\mathcal{V}|, E \to \infty} \mathbb{E}\left[ \mathcal{L}_{sup} \right] = -\frac{1}{B} \sum_{i=1}^{B} \lim_{|\mathcal{V}|, E \to \infty} \mathbb{E}\left[ \log \sum_{c \in S_i} \text{Softmax}(P_i)_c \cdot Q_{ic} \right]$$

$$= -\frac{1}{B} \sum_{i=1}^{B} \mathbb{E}\left[ \lim_{|\mathcal{V}|, E \to \infty} \log \left( \sum_{c \neq y_i^*} \text{Softmax}(P_i)_c \cdot Q_{ic} + (P_i)_{y_i^*} \cdot Q_{i, y_i^*} \right) \right]$$

$$= -\frac{1}{B} \sum_{i=1}^{B} \mathbb{E} \log \left[ 0 + 1 \right] = 0$$

where the last line follows from (18) and (23). $\qquad\square$

## C  Related Work

### C.1  Graph Classification

Graph classification has been widely applied in fields such as molecular property prediction, protein interaction analysis, and social network modeling (Fang et al., 2022; Wang et al., 2021; Ju et al., 2024). Traditional graph kernel methods (Shervashidze et al., 2011) measure structural similarity through subgraph comparisons, but scale poorly to large graphs. Recent advances in Graph Neural Networks (GNNs) (Welling & Kipf, 2016; Hamilton et al., 2017; Veličković et al., 2017; Xu et al., 2018) have shown superior performance by aggregating local neighborhood information and generating graph-level representations via pooling operators (Lee et al., 2019; Gao & Ji, 2019; Lee et al., 2021). Spectral approaches like EigenMLP (Bo et al., 2023) provide an alternative by encoding global graph structure through Fourier-like eigenvalue embeddings. Despite their success, these methods heavily rely on abundant clean labels, which are often unavailable in real-world scenarios due to annotation cost or inherent uncertainty. While self-supervised graph learning methods (You et al., 2020; Luo et al., 2022; Ju et al., 2023a) avoid labels during pre-training, they still require accurate annotations for downstream classification and tend to degrade significantly under label ambiguity. In this paper, we study the problem of partial label graph learning and utilize relational inference with spatial and spectral cues to address the challenge of label noise.

### C.2  Partial Label Learning

Partial label learning (PLL) considers a weak supervision setting where each training instance is annotated with a candidate label set containing only one correct label (Hüllermeier & Beringer, 2005; Cour et al., 2011; Wang et al., 2022; Yan & Guo, 2023). Early approaches treat all candidates equally by averaging losses, but such uniform assumptions often fail under high label ambiguity. Later works focus on label disambiguation, estimating true labels through confidence-based or similarity-driven refinement (Feng & An, 2019; Wang & Zhang, 2022). Recent advances introduce contrastive learning to PLL (Wang et al., 2022), where prototype-instance alignment helps separate correct labels from distractors. However, these methods are mostly designed for Euclidean data such as images or texts. On graph-structured data, PLL remains underexplored Gu et al. (2024); Gao et al. (2024); Gu et al. (2025a). DEER (Gu et al., 2024) is one of the few attempts. It measures semantic distribution divergence between graph views for contrastive learning and leverages posterior-guided soft label correction. Nonetheless, DEER relies on semantic distribution matching and lacks fine-grained structural modeling. GPCD (Gao et al., 2024) introduces graph potential cause discovery to estimate causal subsets for supervision, but suffers from high training complexity and overlooks global spectral cues. In contrast, our proposed PRISM integrates spatial substructure information and spectral characteristics into a unified relational inference framework, enabling more reliable and stable label disambiguation under complex graph structures.

## D  More Experimental Results

We provide more results under the low ambiguity level ($q = 0.1$, $q = 0.02$) in Table 3. Our PRISM consistently achieves the best performance on all datasets, demonstrating its superiority under mild label noise. It may owe to both spatial and spectral cues captured by our PRISM, which play a pivotal role in resolving semantic ambiguities. These results complement the main findings in Section 4.2, and additionally explain the advantage of our framework for keeping strong generalization under different noise regimes.

## E  Details of Baselines

We compare our proposed PRISM against a broad spectrum of baseline models categorized into seven distinct groups: (a) *Graph neural networks*: GCN (Welling & Kipf, 2016), GAT (Veličković et al., 2017), GIN (Xu et al., 2018), and GraphSAGE (Hamilton et al., 2017); (b) *Hierarchical graph pooling*: TopKPool (Gao & Ji, 2019), SAGPool (Lee et al., 2019), EdgePool (Diehl, 2019), and ASAP (Ranjan et al., 2020) (all using GraphSAGE as the encoder backbone); (c) *Graph augmentation method*: Graph Transplant (Park et al., 2022); (d) *Unsupervised contrastive graph learning*:

Table 3: The compared accuracy (mean%±std%) on different graph classification datasets. The best results are highlighted in boldface and the second best results are underlined. $q = P(\overline{y} \in Y | \overline{y} \neq y)$ reflecting the level of label ambiguity.

| Dataset | ENZYMES | Letter-High | COIL-DEL | CIFAR10 | COLORS-3 |
|---|---|---|---|---|---|
| Methods | $q = 0.1$ | $q = 0.1$ | $q = 0.02$ | $q = 0.1$ | $q = 0.1$ |
| GCN | $61.33_{\pm2.85}$ | $50.09_{\pm0.70}$ | $60.77_{\pm1.71}$ | $47.18_{\pm1.09}$ | $90.34_{\pm0.06}$ |
| GAT | $58.22_{\pm3.03}$ | $73.39_{\pm1.41}$ | $69.11_{\pm2.86}$ | $57.56_{\pm0.65}$ | $89.33_{\pm1.42}$ |
| GIN | $59.78_{\pm4.58}$ | $55.83_{\pm4.28}$ | $55.94_{\pm1.69}$ | $47.29_{\pm0.61}$ | $63.01_{\pm1.45}$ |
| GraphSAGE | $60.89_{\pm1.09}$ | $78.20_{\pm1.17}$ | $71.40_{\pm2.15}$ | $57.22_{\pm0.67}$ | $91.70_{\pm2.18}$ |
| TopKPool | $53.11_{\pm4.12}$ | $67.07_{\pm1.60}$ | $55.80_{\pm4.86}$ | $55.26_{\pm0.85}$ | $82.35_{\pm1.36}$ |
| SAGPool | $56.89_{\pm5.37}$ | $67.42_{\pm1.91}$ | $52.94_{\pm2.59}$ | $54.23_{\pm0.53}$ | $76.99_{\pm4.39}$ |
| EdgePool | $58.67_{\pm2.67}$ | $70.49_{\pm3.29}$ | $68.74_{\pm1.85}$ | $55.09_{\pm0.61}$ | $87.47_{\pm0.41}$ |
| ASAP | $60.89_{\pm2.67}$ | $71.25_{\pm1.44}$ | $59.03_{\pm3.09}$ | $54.56_{\pm0.66}$ | $77.84_{\pm1.26}$ |
| Graph Transplant | $61.56_{\pm2.86}$ | $80.75_{\pm0.60}$ | $80.09_{\pm0.75}$ | $56.87_{\pm1.28}$ | $85.48_{\pm0.89}$ |
| PiCO | $61.08_{\pm6.67}$ | $81.27_{\pm1.60}$ | $84.88_{\pm1.09}$ | $57.70_{\pm0.82}$ | $65.68_{\pm1.07}$ |
| TGNN | $62.44_{\pm3.01}$ | $78.55_{\pm0.78}$ | $70.49_{\pm0.87}$ | OOM | $93.16_{\pm1.55}$ |
| GraphCL | $61.78_{\pm1.51}$ | $78.43_{\pm0.85}$ | $78.83_{\pm1.06}$ | $57.62_{\pm0.56}$ | $92.71_{\pm1.61}$ |
| GraphACL | $58.22_{\pm1.51}$ | $81.04_{\pm1.01}$ | $80.66_{\pm0.41}$ | $57.65_{\pm0.21}$ | $92.05_{\pm0.50}$ |
| DEER | $67.11_{\pm1.66}$ | $83.48_{\pm0.92}$ | $87.86_{\pm1.41}$ | $61.45_{\pm0.40}$ | $96.23_{\pm2.94}$ |
| **PRISM (Ours)** | $\mathbf{68.00}_{\pm1.63}$ | $\mathbf{84.87}_{\pm0.74}$ | $\mathbf{89.29}_{\pm1.36}$ | $\mathbf{61.73}_{\pm0.87}$ | $\mathbf{98.36}_{\pm1.07}$ |

GraphCL (You et al., 2020) and GraphACL (Luo et al., 2023); (e) *Semi-supervised graph learning*: TGNN (Ju et al., 2023b); (f) *Partial label learning in vision*: PiCO (Wang et al., 2022), adapted with a GraphSAGE encoder; (g) *Partial label learning for graphs*: DEER (Gu et al., 2024). The details of baselines are presented below.

- **GCN** (Welling & Kipf, 2016): A convolutional network that leverages renormalized adjacency matrices to aggregate neighborhood information in a computationally efficient manner.

- **GAT** (Veličković et al., 2017): It introduces attention weights over neighbors, enabling each node to prioritize important neighbors during message passing.

- **GIN** (Xu et al., 2018): It uses MLPs to approximate injective functions over sets of neighbors, achieving powerful discriminative capacity aligned with the Weisfeiler-Lehman test.

- **GraphSAGE** (Hamilton et al., 2017): It aggregates information from randomly sampled neighbors, making it scalable to large graphs.

- **TopKPool** (Gao & Ji, 2019): It selects top-ranked nodes according to an adaptive projection score to form a pooled graph with reduced size.

- **SAGPool** (Lee et al., 2019): It utilizes self-attention scores computed from graph convolutions to guide node selection for pooling, capturing both feature and structural signals.

- **EdgePool** (Diehl, 2019): It contracts edges iteratively to coarsen the graph while maintaining crucial topological structures.

- **ASAP** (Ranjan et al., 2020): It combines node selection and clustering by learning soft assignments over local $h$-hop neighborhoods, allowing for adaptive structure-aware pooling.

- **Graph Transplant** (Park et al., 2022): A mixup-inspired data augmentation strategy that extracts meaningful subgraphs based on node saliency and generates hybrid samples through substructure-level interpolation.

- **GraphCL** (You et al., 2020): A contrastive learning framework that maximizes agreement between different augmented views of a graph, using stochastic transformations.

- **GraphACL** (Luo et al., 2023): It improves contrastive representation learning by constructing adversarial hard negatives and regularizing the feature space with orthogonality and divergence constraints.

- **TGNN** (Ju et al., 2023b): A dual-view semi-supervised framework that integrates message passing and kernel-based reasoning, encouraging consistency across views to exploit both labeled and unlabeled graphs.

- **PiCO** (Wang et al., 2022): It learns a set of class-wise prototypes and employs contrastive objectives to align instance embeddings with their correct prototypes. We adopt a GraphSAGE encoder to enable graph-level applications.
- **DEER** (Gu et al., 2024): A partial-label graph learning method that selects reliable positive pairs by measuring distribution divergence across augmented views. It also performs soft label correction via posterior estimation. However, it does not explicitly model substructures or spectral signals, limiting its granularity in structural reasoning.

**Training Protocol.** All baselines are trained under partial-label supervision using a cross-entropy loss over candidate label sets. TGNN additionally optimizes a consistency loss between dual views. For contrastive methods such as GraphCL, GraphACL, and PiCO, we apply their contrastive loss and the cross-entropy loss. Pooling-based methods are evaluated with a fixed pooling ratio of $0.6$ and use GraphSAGE as their encoder. All models are trained under identical random seeds for fair comparisons.

## F    DETAILS OF IMPLEMENTATION

We implement all models using PyTorch with the PyG backend. A two-layer GraphSAGE with 512 hidden units is adopted as the backbone encoder. The EMA momentum for prototype and label updates is set to $m = \beta = 0.99$, and the propagation momentum is $\mu = 0.9$ with $E = 2$ steps. Eigen-decomposition is precomputed via sparse solvers to ensure efficiency. For ENZYMES, Letter-High, COIL-DEL, and COLORS-3, we partition the data into training, validation, and test sets with a ratio of $80\%:5\%:15\%$. For CIFAR10, we follow the popular split of $45,000$ training, $5,000$ validation, and $10,000$ test samples, consistent with the protocol in (Dwivedi et al., 2020). All these results are averaged over five independent runs with individual seeds.

