# OpenReview forum: "PRISM: Partial-label Relational Inference with Spatial and Spectral Cues"
_ICLR.cc/2026/Conference — ICLR 2026 Poster_

### Official Review · Reviewer_Nyo7 · 2025-10-28

**Soundness:** 2
**Presentation:** 2
**Contribution:** 2
**Rating:** 4
**Confidence:** 3

**Summary:**

This paper proposes PRISM, a framework for Partial-Label Graph Learning (PLGL). It resolves label ambiguity by integrating local spatial cues (substructure matching) with global spectral cues (frequency analysis). These views are used to construct a relational graph that guides an iterative label propagation process. Experiments show PRISM achieves state-of-the-art results, demonstrating strong robustness to label noise.

**Strengths:**

1. The manuscript tackles an important, practical task: learning from graphs that only provide a candidate label set, and it states the setup clearly for real applications.
2. It shows strong results with noticeable margins on several benchmarks and stays stable as label noise increases, which points to solid robustness.
3. The pipeline is modular and easy to adapt and swap encoders or spectrum modules without redesigning the whole method.

**Weaknesses:**

1. The theory relies on very strong assumptions and does not explain how the model actually learns under noisy supervision; it mostly describes behavior at an ideal fixed point.
2. The paper does not convincingly show that the two views truly help each other; ablations suggest the label-propagation step may account for most gains, not the new encoders, so the claimed synergy remains unclear.
3. Some key implementation details are missing, such as how confidence-based filtering is done, and the complexity claim skips the likely quadratic cost of cross-graph similarity and Top-k selection, which can mislead readers about scalability.

**Questions:**

1. Theorems 1/2 assume the GNN already maps samples near the correct prototypes and the classifier recognizes them perfectly; they describe an ideal fixed point rather than how the model reaches it under noisy supervision, offering little insight into the core challenge.
2. Is the claim that the model's complexity is O(|E| d) reasonable? Finding the top-k neighbors for all graphs in a batch seems like it would be at least an O(N^2) operation, which can be very slow. The paper doesn't seem to have accounted for this cost.
3. The paper claims the spatial and spectral views are complementary, but neither the theory nor the experiments really show how they help each other. Is there an example where one view gets it wrong, but the other one corrects it? Otherwise, it looks more like a simple combination of two methods.
4. Looking at your ablation study (Table 2), removing the "Relational Inference" step causes the biggest performance drop. This makes me wonder: is the final label propagation step the real key to the performance boost? It might mean that the spatial and spectral encoders you focus on are not the main reason for the model's great results.
5. In Section 3.1, you mentioned a key step called "confidence-based filtering" for updating the prototypes, but it's never explained how it works. Without it, it's hard to fully understand your method.

---

> ### Author Response · Authors · 2025-12-03
> **Response to Reviewer Nyo7 (I)**
>
> We are truly grateful for the time you have taken to review our paper and your insightful review. Here we address your comments in the following.
>
>
> > **Q1**: The theory relies on very strong assumptions and does not explain how the model actually learns under noisy supervision; it mostly describes behavior at an ideal fixed point.
>
> **A1**: Thanks to the reviewer for pointing out that the conditions for our theorem are too strong. The original Theorem 2 assumes that the classifier can correctly identify the true label from the true class “prototype”, since the true prototype is unknown (albeit fixed in the underlying data-generating process), the assumption that a classifier can perfectly reconstruct the true label from this prototype is indeed rather strong. To relax this, we introduce a refined result, stated as Theorem 3, which makes no assumption on the classifier itself and instead only requires independence of training samples within each mini-batch. More details and proof about Theorem 3 can be seen at the answer of Question 4.
>
>
> > **Q2**: The paper does not convincingly show that the two views truly help each other; ablations suggest the label-propagation step may account for most gains, not the new encoders, so the claimed synergy remains unclear.
>
> **A2**: We appreciate the concern and agree that the current draft could better articulate how the spatial and spectral views help each other, beyond the relational inference step. Conceptually, the synergy comes from the fact that the relational graph used for label propagation is constructed from both encoders: the spatial adjacency A^{spa} arises from prototype-guided substructure similarity (Eq. (5)), while the spectral adjacency A^{spe} comes from multi-band frequency similarity (Eq. (10)). If either view is weak, another view has the opportunity to make a correction to it, thus praying for a robust effect. Empirically, Table 2 already shows that both views matter: removing the spatial branch (“w/o Spa”) or the spectral branch (“w/o Spe”) consistently hurts performance on all three datasets and noise levels, and both single-view variants are clearly worse than the full two-view model. To make this clearer, in the revision we will explicitly highlight that “w/o Rel. Infer.” still uses both encoders, so the gap between this variant and the full model measures the extra benefit of synergy through relational propagation.
>
> > **Q3**: Some key implementation details are missing, such as how confidence-based filtering is done, and the complexity claim skips the likely quadratic cost of cross-graph similarity and Top-k selection, which can mislead readers about scalability.
>
> **A3**: Thanks for your comment. The reviewer is right that Sec. 3.5 currently focuses on the per-graph encoding cost and does not spell out the cost of constructing the cross-graph relational graphs, which can be misleading. We will clarify this in the revision. In terms of complexity, the cross-graph similarity + Top-k cost per training step is O( |P_B| \cdot F) + O(|P_B| \codt \log k), where |P_B|  is the number of graph pairs in the batch with S_i ∩ S_j \neq ∅, and F is the feature dimension of the substructure/spectral embeddings. In the worst case |P_B| = O(B^2), but in partial-label settings with modest candidate set sizes this is substantially smaller.

---

> ### Author Response · Authors · 2025-12-03
> **Response to Reviewer Nyo7 (II)**
>
> > **Q4**: Theorems 1/2 assume the GNN already maps samples near the correct prototypes and the classifier recognizes them perfectly; they describe an ideal fixed point rather than how the model reaches it under noisy supervision, offering little insight into the core challenge.
>
> **A4**: Thanks for your comment. To relax this concern, we introduce a refined result, stated as Theorem 3:
> ```
> (Theorem 3) Assume data $(X_i)_{i \in B}$ are independent from each other in each mini-batch with batch size $B$. Then,
>     \begin{equation}
>     \label{cls3_1}
>         Q_i \xrightarrow{a.s.} Y_i^*
>     \end{equation}
>     as $|\mathcal{V}_i|\to \infty,T \to \infty$. Moreover, for any $\delta > 0$, with at least probability $1 - \delta$,
>     \begin{equation}
>     \label{cls3_2}
>         \mathcal{L}_{sup}^{(o)}(T)  \leq \left|\mathcal{C}\right| \left\{ \left[\frac{1}{2\min_{1\leq c \leq |\mathcal{C}| }|\mathcal{S}_c^*|  } \log\left(\frac{|\mathcal{C}|}{\delta}\right) \right]^{1/2} +  \min_{1 \leq c \leq |\mathcal{C}|}\mathcal{M}_c^{(o)}(T)  \right\}, \quad o \in \{spa,spe \},
>     \end{equation}
>
> where $\mathcal{M}_c^{(o)}(T) = \mathbb{E}\left[\left(-\log\right)\left(\text{softmax}(P_i^{(o)}(T))_c \right)| y_i^* = c\right]$ and $|\mathcal{S}_c^*|$ denotes the number of samples in mini-batch with true label $c$.
> \textbf{Proof of Theorem 3}:
>     By Theorem 1 and formula (13), (15) we obtain
>     \[Y_i^{(T)} \xrightarrow{a.s.} Y_i^*,\quad  Q_i \xrightarrow{a.s.} Y_i^* \]
>     as $T \to \infty$. For the sake of convenience, we omit the parameter $T$ of the notations $P_i^{(o)}, Q_{i}$ in the following proof, which are actually the soft label confidence matrix and MLP-based classifier output after the $T$-th iteration. Since $B = \sum_{c=1}^{|\mathcal{C}|} |S_c^*|$, by Jensen's inequality we have
>
>     \begin{equation}
>         \begin{aligned}
>             \mathcal{L}_{sup}^{(o)}(T) &= - \frac{1}{B} \sum_{i=1}^{B} \log \sum_{c \in \mathcal{S}_i} \text{Softmax}(\bm{P}_i^{(o)})_c \cdot \bm{Q}_{ic} \\
>             % &\leq -\sum_{c=1}^{|\mathcal{C}|} \frac{1}{|S_c^*|} \log\sum_{i \in S_c^*} \text{Softmax}(\bm{P}_i^{(o)})_c \cdot \bm{Q}_{ic} \\
>             &\leq \sum_{c=1}^{|\mathcal{C}|} \frac{1}{|S_c^*|} \sum_{i \in S_c^*} (-\log) \big[\sum_{\tilde{c} \in \mathcal{S}_i}\text{Softmax}(\bm{P}_i^{(o)})_{\tilde{c}} \cdot \bm{Q}_{i\tilde{c}}\big],
>         \end{aligned}
>         \nonumber
>     \end{equation}
>     Consider $\mathbb{P}(\mathcal{L}_{sup}^{(o)}(T) > \epsilon)$, by careful partition of the probability space we obtain
>     \begin{equation}
>         \begin{aligned}
>             \mathbb{P}(\mathcal{L}_{sup}^{(o)}(T) > \epsilon) &\leq \mathbb{P}\left( \sum_{c=1}^{|\mathcal{C}|} \frac{1}{|S_c^*|} \sum_{i \in S_c^*} (-\log) \big[\sum_{\tilde{c} \in \mathcal{S}_i}\text{Softmax}(\bm{P}_i^{(o)})_{\tilde{c}} \cdot \bm{Q}_{i\tilde{c}}\big] > \epsilon \right) \\
>             &\leq \mathbb{P}\left( \bigcup_{c=1}^{|\mathcal{C}|}  \left\{\frac{1}{|S_c^*|} \sum_{i \in S_c^*} (-\log) \big[\sum_{\tilde{c} \in \mathcal{S}_i}\text{Softmax}(\bm{P}_i^{(o)})_{\tilde{c}} \cdot \bm{Q}_{i\tilde{c}}\big] > \frac{\epsilon}{|\mathcal{C}|} \right\} \right) \\
>             &\leq \sum_{c=1}^{|\mathcal{C}|} \mathbb{P}\left( \frac{1}{|S_c^*|} \sum_{i \in S_c^*} (-\log) \big[\sum_{\tilde{c} \in \mathcal{S}_i}\text{Softmax}(\bm{P}_i^{(o)})_{\tilde{c}} \cdot \bm{Q}_{i\tilde{c}}\big] > \frac{\epsilon}{|\mathcal{C}|} \right) \\
>             &\leq \sum_{c=1}^{|\mathcal{C}|} \mathbb{P}\left( \frac{1}{|S_c^*|} \sum_{i \in S_c^*} (-\log) \big[\sum_{\tilde{c} \in \mathcal{S}_i}\text{Softmax}(\bm{P}_i^{(o)})_{\tilde{c}} \cdot \bm{Q}_{i\tilde{c}}\big] > \frac{\epsilon}{|\mathcal{C}|} \right) \\
>             &\leq \sum_{c=1}^{|\mathcal{C}|} \mathbb{P}\left( \frac{1}{|S_c^*|} \sum_{i \in S_c^*} (-\log) \big[\text{Softmax}(\bm{P}_i^{(o)})_{c} \big] > \frac{\epsilon}{|\mathcal{C}|} \right), \\
>         \end{aligned}
>         \nonumber
>     \end{equation}
> ```
>  As $T \to \infty$ where the last comes from (\ref{cls3_1}) and $Y_i^*$ is the one-hot true label vector with only the $j$-th component equal to 1.

---

> ### Author Response · Authors · 2025-12-03
> **Response to Reviewer Nyo7 (III)**
>
> Then followed by Hoeffiding's inequality, notice $(X_i)_{i \in B}$ are independent with each other in each mini-batch and recall the definition of $\mathcal{M}_c^{(o)}(T)$, we have
> ```
>     \begin{equation}
>         \begin{aligned}
>             \mathbb{P}(\mathcal{L}_{sup}^{(o)}(T) > \epsilon) &\leq \sum_{c=1}^{|\mathcal{C}|} \mathbb{P}\left( \frac{1}{|S_c^*|} \sum_{i \in S_c^*} (-\log) \big[\text{Softmax}(\bm{P}_i^{(o)})_{c} \big] - \mathcal{M}_c^{(o)}(T) > \frac{\epsilon}{|\mathcal{C}|} - \mathcal{M}_c^{(o)}(T) \right) \\
>             &\leq \sum_{c=1}^{|\mathcal{C}|} \exp \left\{ -2|S_c^*|\left( \frac{\epsilon}{|\mathcal{C}|} - \mathcal{M}_c^{(o)}(T) \right)^2 \right\}\\
>             &\leq |\mathcal{C}|\exp \left\{ -2\left(\min_{1\leq c \leq |\mathcal{C}|}|S_c^*|\right)\left( \frac{\epsilon}{|\mathcal{C}|} - \min_{1\leq c \leq |\mathcal{C}|}\mathcal{M}_c^{(o)}(T)\right)^2 \right\}\\
>             &\leq \delta
>         \end{aligned}
>         \nonumber
>     \end{equation}
>     when $\epsilon  > |\mathcal{C}| \left\{ \left[\frac{1}{2\min_{1\leq c \leq |\mathcal{C}| }|\mathcal{S}_c^*|  } \log\left(\frac{|\mathcal{C}|}{\delta}\right) \right]^{1/2} +  \min_{1 \leq c \leq |\mathcal{C}|}\mathcal{M}_c^{(o)}(T)\right\}$, which completes the proof of (\ref{cls3_2}).
> ```
>
> Theorem 3 provides a generalization bound for the loss function. It shows that the upper bound decreases as the training time T increases, and grows with the number of classes |\mathcal{C}|, indicating that the learning problem becomes intrinsically harder as the label space enlarges. In the ideal case, the classifier output softmax(P_i^{(o)}(T))_{y^*_i} should converge to 1 as T \to \infty, which implies \mathcal{M}_c^{(o)}(T) \to 0. Consequently, the generalization bound becomes progressively tighter with longer training. Moreover, the assumption of independence of samples within each mini-batch is relatively mild and well-justified in practice, especially when the mini-batch size is small.
>
>
> > **Q5**: Is the claim that the model's complexity is O(|E| d) reasonable? Finding the top-k neighbors for all graphs in a batch seems like it would be at least an O(N^2) operation, which can be very slow. The paper doesn't seem to have accounted for this cost.
>
> **A5**: Thanks for your comment. Thank the reviewer for mentioning the specific details of complexity calculation. From a theoretical perspective, The per-graph encoding cost is as in Sec. 3.5: Spatial GNN is O(L |E| d), spectral branch based on truncation preprocessing eigenvectors once is O(T |E| d), or equivalently O(T N d) or sparse graphs with bounded degree); the cross-graph similarity + Top-k cost per training step is O( |P_B| \cdot F) + O(|P_B| \codt \log k) as mentioned in the answer of Question 3, so strictly speaking, the total complexity is O(L ∣E∣ d + T ∣E∣ d + ∣P B∣ (F + logk)). Since L,T,F, and k are small constants and B is bounded (mini-batch training), the asymptotic dependence on the graph structure remains linear in ∣E∣, but the reviewer is correct that we should explicitly mention the O(∣P B∣) term rather than collapsing everything into O(∣E∣d). From an experimental perspective, reviewer can see the following running time comparison chart of different methods, we will add the content above in the revised version.
>
> |Method|CIFAR10||
> |-|-|-|
> ||Time (min)| Acc (%)|
> |EdgePool|276.19|50.17|
> |Graph Transplant|169.35|53.79|
> |PRISM|155.51|58.29|
>
>
> > **Q6**: The paper claims the spatial and spectral views are complementary, but neither the theory nor the experiments really show how they help each other. Is there an example where one view gets it wrong, but the other one corrects it? Otherwise, it looks more like a simple combination of two methods.
>
> **A6**: Thanks for your comment. The current text only qualitatively states that the views are complementary; we agree that more concrete evidence would strengthen this claim. In practice, the two views capture different failure modes: The spatial view is strong when class-discriminative patterns are localized (e.g., small motifs in ENZYMES), but can be confused when graphs share similar local motifs yet differ in global organization; the spectral view is robust to such global differences, since different frequency bands emphasize different scales of smoothness/irregularity, but it can be less sensitive to subtle local motif changes when the overall topology is similar. Relational inference combines these two signals: a graph that is ambiguous in one view (few high-similarity neighbors) can still obtain reliable neighbors from the other view, and Eq. (13) fuses both adjacencies before propagation, which improves the robustness of our method across different datasets. In the revision, we will add another qualitative case study, which will make the “complementary” role of the two views more concrete.

---

> ### Author Response · Authors · 2025-12-03
> **Response to Reviewer Nyo7 (IV)**
>
> > **Q7**: Looking at your ablation study (Table 2), removing the "Relational Inference" step causes the biggest performance drop. This makes me wonder: is the final label propagation step the real key to the performance boost? It might mean that the spatial and spectral encoders you focus on are not the main reason for the model's great results.
>
> **A7**: Thank you for pointing out the strong effect of the “Relational Inference” ablation in Table 2. We agree that this deserves a more careful discussion. The ablation results support three parts of PRISM: PRISM w/o Rel. Infer” keeps both encoders but removes label propagation, since it already reaches performance comparable to or slightly below the best prior method (e.g., ENZYMES with q=0.3), which suggests the encoders themselves are competitive partial-label learners; removing either encoder (“w/o Spa” or “w/o Spe”) consistently degrades accuracy relative to the full model, indicating that both encoders contribute non-redundant information; the additional gain of 3–6% absolute accuracy from adding relational inference reflects how much better we can do once we have strong spatial and spectral views and can propagate labels on a graph-of-graphs. Therefore, the spatial and spectral encoders should have contributed to the model's great results significantly.
>
> > **Q8**: In Section 3.1, you mentioned a key step called "confidence-based filtering" for updating the prototypes, but it's never explained how it works. Without it, it's hard to fully understand your method.
>
> **A8**: Thanks for your comment. The “confidence-based filtering” mentioned under Eq. (3) governs which graphs contribute to the prototype update for each class. Concretely, we define B_c = { i | c \in S_i, Q_{i,c} \geq \tau } with a fixed threshold τ. Intuitively, this means only graphs that both (i) include c in their candidate set and (ii) currently have sufficiently high soft confidence on c are allowed to “pull” the class prototype toward their embedding. This filtering prevents noisy early predictions from dominating the prototypes and ensures that prototypes gradually concentrate around graphs that are consistently supported by relational inference. In the revision, we will Explicitly add the definition of B_c and the threshold τ in Sec. 3.1 after Eq. (3), and  briefly discuss how this mechanism interacts with the EMA update of Q to form a stable closed loop.
>
>
> If there are any additional notable points of concern that we have not yet addressed, please do not hesitate to share them, and we will promptly attend to those points.

---

### Official Review · Reviewer_Qmcu · 2025-10-30

**Soundness:** 2
**Presentation:** 3
**Contribution:** 2
**Rating:** 6
**Confidence:** 3

**Summary:**

The paper tackles the practical and underexplored problem of Partial-label Graph Learning (PLGL), which reflects real-world situations where graph labels are incomplete or uncertain. The proposed method PRISM integrates three complementary components, i.e., prototype-guided substructure alignment, spectral modeling, and hybrid relational graph propagation, to address label ambiguity from spatial, spectral, and relational perspectives.

**Strengths:**

The approach is conceptually coherent and well-motivated, combining local structural cues with global spectral information for more reliable supervision. Experimental results on multiple benchmarks demonstrate clear performance improvements over existing weakly supervised and graph learning methods, supporting the effectiveness of the proposed framework.

**Weaknesses:**

1. Graphs differ significantly in size and structure, making cross-graph substructure alignment both conceptually unclear and potentially computationally expensive. The paper does not explain how this process is implemented or efficiently approximated, and when dealing with large graphs with many nodes and edges, it could still lead to severe computational bottlenecks.

2. Each graph has its own Laplacian basis, making frequency bands difficult to compare across graphs. The paper does not explain how spectral features are aligned or normalized, and when the structural differences between graphs are large, ensuring that their spectral representations are meaningfully aligned remains a significant challenge.

3. The theory assumes a perfectly trained classifier with one-hot outputs. In practice, momentum updates and label propagation interact, so this assumption may not hold.

4. Limited robustness evaluation. The work does not systematically evaluate different candidate set sizes, nor does it examine performance under varying noise levels or open-set scenarios.

**Questions:**

1. How is the cross-graph substructure alignment implemented in practice? Can the authors show that this step does not cause extra computational cost?
2. Since each graph has its own Laplacian basis, how does the method handle spectral inconsistency when using multi-band features?
3. The convergence proof assumes a well-trained classifier with correct one-hot outputs. What happens if this assumption does not hold? Any discussion or experiments?
4. Could the authors test the model under different candidate set sizes, noise levels, or open-set cases (where true labels are missing)?

---

> ### Author Response · Authors · 2025-12-03
> **Response to Reviewer Qmcu (I)**
>
> We are truly grateful for the time you have taken to review our paper, your insightful comments and support. Your positive feedback is incredibly encouraging for us! In the following response, we would like to address your major concern and provide additional clarification.
>
> > **Q1**: Graphs differ significantly in size and structure, making cross-graph substructure alignment both conceptually unclear and potentially computationally expensive. The paper does not explain how this process is implemented or efficiently approximated, and when dealing with large graphs with many nodes and edges, it could still lead to severe computational bottlenecks.
>
> **A1**: Thanks for your comment. Conceptually, PRISM does not perform any combinatorial subgraph isomorphism or node-to-node matching across graphs. Instead, we learn a prototype bank p_c, compute class-aware embeddings r_i^{(c)}, and define prototype-aware substructure similarity which refers to Eq. (5). for any pair of graphs (G_i,G_j) that share at least one candidate label. Thus, “cross-graph substructure alignment” in PRISM means aligning prototype-conditioned substructure embeddings in a shared feature space, not matching raw graph structures. This makes the notion of alignment well-defined even when graphs differ greatly in size and topology. As for the computational concern, the cost of building and using A^{spa} is dominated by the base GNN and does not change the overall asymptotic complexity claimed in Sec. 3.5, while for the cost of building spectral encoder, by Eq. (6) we can see the eigenvalue computation is truncated and applied to sparse Laplacians, which makes the overhead modest compared to standard GNN training. From an experimental perspective, reviewer can see the following running time comparison chart of different methods.
>
> |Method|CIFAR10||
> |-|-|-|
> ||Time (min)| Acc (%)|
> |EdgePool|276.19|50.17|
> |Graph Transplant|169.35|53.79|
> |PRISM|155.51|58.29|
>
> > **Q2**: Each graph has its own Laplacian basis, making frequency bands difficult to compare across graphs. The paper does not explain how spectral features are aligned or normalized, and when the structural differences between graphs are large, ensuring that their spectral representations are meaningfully aligned remains a significant challenge.
>
> **A2**: Thanks for your comment. The reviewer is right that each graph has its own Laplacian basis. PRISM does not assume that eigenvectors are directly comparable across graphs. Instead, it achieves alignment in this ways: Normalized Laplacian & harmonic embedding of eigenvalues(Eq. (6)), Band-specific node and graph embeddings with shared encoders(Eq. (7)-(8)), Band-wise similarity(not eigenvector-wise matching, Eq. (10)). Finally, Theorem 1 shows that under mild assumptions, the probabilities of forming edges in both between two graphs with the same true label converge to 1 as graph sizes increase (Eqs. (11)–(12)). This provides a theoretical justification that the spectral similarity we construct is label-consistent across graphs despite basis differences. Thanks for the reviewer's question and we will highlight this alignment mechanism and the role of normalized spectrum + harmonic embedding more explicitly in the revision.
>
>
> > **Q3**: The theory assumes a perfectly trained classifier with one-hot outputs. In practice, momentum updates and label propagation interact, so this assumption may not hold.
>
> **A3**: Thanks for your comment. The original Theorem 2 assumes that the classifier can identify the true label from the true "prototype" of each class, and that the latent variables $z_i$ of each sample $i$ are guaranteed by Theorem 1 to asymptotically reach the prototype of $y_i^*$ under sufficient sample size and learning. Of course, the true "prototype" is always unknown and present, so the assumption that the classifier can reconstruct the true label from the "prototype" is indeed quite stringent. To address this, we present a refined Theorem 2 (or Theorem 3), which, requires no assumptions about the classifier, and only need the assumption of independence of training samples within each mini-batch. More details and proof towards this Theorem 3 can be seen on the answer of Question 7.

---

> ### Author Response · Authors · 2025-12-03
> **Response to Reviewer Qmcu (II)**
>
> > **Q4**: Limited robustness evaluation. The work does not systematically evaluate different candidate set sizes, nor does it examine performance under varying noise levels or open-set scenarios.
>
> **A4**: Thanks for your comment. We follow Gu et al. (2024) to generate partial labels: each incorrect label is included in the candidate set with probability q = Prob(y \in \mathcal{Y} | y \neq y^*), which jointly controls noise level and expected candidate set size. Table 1 reports results across multiple q values which already gives a nontrivial picture of robustness w.r.t. both noise level and candidate-set cardinality. In addition, Sec. 4.4 runs a sensitivity analysis over the sparsity levels k_a, k_e of the spatial and spectral relational graphs, showing that performance is stable over a broad range of neighbor counts (1–5) and only degrades mildly when the relational graph becomes overly dense. As for open-set partial-label scenarios, our current work follows the standard partial-label learning assumption that the true label is always contained in the candidate set (as in prior PLL works and the data generation protocol we adopt), open-set PLL is an important but orthogonal problem and is not covered by our theoretical analysis at present.
>
>
> > **Q5**: How is the cross-graph substructure alignment implemented in practice? Can the authors show that this step does not cause extra computational cost?
>
> **A5**: Thanks for your comment. Since all cross-graph operations are performed on graph-level vectors (substructure embeddings and band-level spectral embeddings), not on nodes or subgraphs, the alignment step is light-weight compared to GNN message passing and eigen-decomposition. This is why the overall complexity remains O(|E|d), and in practice the additional wall-clock overhead is modest. From an experimental perspective, we will report runtime comparisons to a plain GNN baseline and justify this step does not cause extra computational cost in the revised version.
>
>
> > **Q6**: Since each graph has its own Laplacian basis, how does the method handle spectral inconsistency when using multi-band features?
>
> **A6**: Thanks for your comment. Multi-band features in PRISM are built on normalized Laplacian eigenpairs and shared harmonic encoders, which means the notion of “low” vs “high” frequency is intrinsically comparable across graphs(normalized spectrum in [0,2] and shared mapping ρ(λ)). Band similarity across graphs is defined via cosine similarity of these learned embeddings, taking the maximum over bands(Eq. (10)), so we do band-to-band matching in the learned latent space, not raw eigenvector matching. The eigengap assumption in Theorem 1 ensures that the leading eigenspaces are stable under perturbations of the adjacency, which, together with the label-dependent adjacency assumption, guarantees that A^{spe} tends to connect same-label graphs with high probability (Eq. (12)). This gives a theoretical handle on spectral “consistency” despite different Laplacian bases.

---

> ### Author Response · Authors · 2025-12-03
> **Response to Reviewer Qmcu (III)**
>
> > **Q7**: The convergence proof assumes a well-trained classifier with correct one-hot outputs. What happens if this assumption does not hold? Any discussion or experiments?
>
> **A7**: Thanks for your comment. We give a new Theorem 3 as follows, which provides a generalization bound for the loss function. It reveals that the upper bound decreases with increasing training time $T$ and increases with increasing number of label classes $|\mathcal{C}|$, meaning that the learning curve widens as the number of label candidates increases. Notice that under ideal situation, the classifier output $\text{softmax}(P_i^{(o)}(T))_{y_i^*}$ should tend to 1 as $T \to \infty$, which leads to $\mathcal{M}_c^{(o)}(T) \to 0$. Thus, the generalization bound would be tighter as we train longer. What's more, the assumption of sample independence within each mini-batch is relatively relaxed and natural when the mini-batch size is small.
>
> ```
> (Theorem 3) Assume data $(X_i)_{i \in B}$ are independent from each other in each mini-batch with batch size $B$. Then,
>     \begin{equation}
>     \label{cls3_1}
>         Q_i \xrightarrow{a.s.} Y_i^*
>     \end{equation}
>     as $|\mathcal{V}_i|\to \infty,T \to \infty$. Moreover, for any $\delta > 0$, with at least probability $1 - \delta$,
>     \begin{equation}
>     \label{cls3_2}
>         \mathcal{L}_{sup}^{(o)}(T)  \leq \left|\mathcal{C}\right| \left\{ \left[\frac{1}{2\min_{1\leq c \leq |\mathcal{C}| }|\mathcal{S}_c^*|  } \log\left(\frac{|\mathcal{C}|}{\delta}\right) \right]^{1/2} +  \min_{1 \leq c \leq |\mathcal{C}|}\mathcal{M}_c^{(o)}(T)  \right\}, \quad o \in \{spa,spe \},
>     \end{equation}
>     where $\mathcal{M}_c^{(o)}(T) = \mathbb{E}\left[\left(-\log\right)\left(\text{softmax}(P_i^{(o)}(T))_c \right)| y_i^* = c\right]$ and $|\mathcal{S}_c^*|$ denotes the number of samples in mini-batch with true label $c$.
> \textbf{Proof of Theorem 3}:
>     By Theorem 1 and formula (13), (15) we obtain
>     \[Y_i^{(T)} \xrightarrow{a.s.} Y_i^*,\quad  Q_i \xrightarrow{a.s.} Y_i^* \]
>     as $T \to \infty$.
> ```
> For the sake of convenience, we omit the parameter $T$ of the notations $P_i^{(o)}, Q_{i}$ in the following proof, which are actually the soft label confidence matrix and MLP-based classifier output after the $T$-th iteration. Since $B = \sum_{c=1}^{|\mathcal{C}|} |S_c^*|$, by Jensen's inequality we have
> ```
> \begin{equation}
>         \begin{aligned}
>             \mathcal{L}_{sup}^{(o)}(T) &= - \frac{1}{B} \sum_{i=1}^{B} \log \sum_{c \in \mathcal{S}_i} \text{Softmax}(\bm{P}_i^{(o)})_c \cdot \bm{Q}_{ic} \\
>             % &\leq -\sum_{c=1}^{|\mathcal{C}|} \frac{1}{|S_c^*|} \log\sum_{i \in S_c^*} \text{Softmax}(\bm{P}_i^{(o)})_c \cdot \bm{Q}_{ic} \\
>             &\leq \sum_{c=1}^{|\mathcal{C}|} \frac{1}{|S_c^*|} \sum_{i \in S_c^*} (-\log) \big[\sum_{\tilde{c} \in \mathcal{S}_i}\text{Softmax}(\bm{P}_i^{(o)})_{\tilde{c}} \cdot \bm{Q}_{i\tilde{c}}\big],
>         \end{aligned}
>         \nonumber
>     \end{equation}
> ```

---

> ### Author Response · Authors · 2025-12-03
> **Response to Reviewer Qmcu (IV)**
>
> Consider $\mathbb{P}(\mathcal{L}_{sup}^{(o)}(T) > \epsilon)$, by careful partition of the probability space we obtain
> ```
> \begin{equation}
>         \begin{aligned}
>             \mathbb{P}(\mathcal{L}_{sup}^{(o)}(T) > \epsilon) &\leq \mathbb{P}\left( \sum_{c=1}^{|\mathcal{C}|} \frac{1}{|S_c^*|} \sum_{i \in S_c^*} (-\log) \big[\sum_{\tilde{c} \in \mathcal{S}_i}\text{Softmax}(\bm{P}_i^{(o)})_{\tilde{c}} \cdot \bm{Q}_{i\tilde{c}}\big] > \epsilon \right) \\
>             &\leq \mathbb{P}\left( \bigcup_{c=1}^{|\mathcal{C}|}  \left\{\frac{1}{|S_c^*|} \sum_{i \in S_c^*} (-\log) \big[\sum_{\tilde{c} \in \mathcal{S}_i}\text{Softmax}(\bm{P}_i^{(o)})_{\tilde{c}} \cdot \bm{Q}_{i\tilde{c}}\big] > \frac{\epsilon}{|\mathcal{C}|} \right\} \right) \\
>             &\leq \sum_{c=1}^{|\mathcal{C}|} \mathbb{P}\left( \frac{1}{|S_c^*|} \sum_{i \in S_c^*} (-\log) \big[\sum_{\tilde{c} \in \mathcal{S}_i}\text{Softmax}(\bm{P}_i^{(o)})_{\tilde{c}} \cdot \bm{Q}_{i\tilde{c}}\big] > \frac{\epsilon}{|\mathcal{C}|} \right) \\
>             &\leq \sum_{c=1}^{|\mathcal{C}|} \mathbb{P}\left( \frac{1}{|S_c^*|} \sum_{i \in S_c^*} (-\log) \big[\sum_{\tilde{c} \in \mathcal{S}_i}\text{Softmax}(\bm{P}_i^{(o)})_{\tilde{c}} \cdot \bm{Q}_{i\tilde{c}}\big] > \frac{\epsilon}{|\mathcal{C}|} \right) \\
>             &\leq \sum_{c=1}^{|\mathcal{C}|} \mathbb{P}\left( \frac{1}{|S_c^*|} \sum_{i \in S_c^*} (-\log) \big[\text{Softmax}(\bm{P}_i^{(o)})_{c} \big] > \frac{\epsilon}{|\mathcal{C}|} \right), \\
>         \end{aligned}
>         \nonumber
>     \end{equation}
> ```
>  As $T \to \infty$ where the last comes from (\ref{cls3_1}) and $Y_i^*$ is the one-hot true label vector with only the $j$-th component equal to 1. Then followed by Hoeffiding's inequality, notice $(X_i)_{i \in B}$ are independent with each other in each mini-batch and recall the definition of $\mathcal{M}_c^{(o)}(T)$, we have
> ```
>     \begin{equation}
>         \begin{aligned}
>             \mathbb{P}(\mathcal{L}_{sup}^{(o)}(T) > \epsilon) &\leq \sum_{c=1}^{|\mathcal{C}|} \mathbb{P}\left( \frac{1}{|S_c^*|} \sum_{i \in S_c^*} (-\log) \big[\text{Softmax}(\bm{P}_i^{(o)})_{c} \big] - \mathcal{M}_c^{(o)}(T) > \frac{\epsilon}{|\mathcal{C}|} - \mathcal{M}_c^{(o)}(T) \right) \\
>             &\leq \sum_{c=1}^{|\mathcal{C}|} \exp \left\{ -2|S_c^*|\left( \frac{\epsilon}{|\mathcal{C}|} - \mathcal{M}_c^{(o)}(T) \right)^2 \right\}\\
>             &\leq |\mathcal{C}|\exp \left\{ -2\left(\min_{1\leq c \leq |\mathcal{C}|}|S_c^*|\right)\left( \frac{\epsilon}{|\mathcal{C}|} - \min_{1\leq c \leq |\mathcal{C}|}\mathcal{M}_c^{(o)}(T)\right)^2 \right\}\\
>             &\leq \delta
>         \end{aligned}
>         \nonumber
>     \end{equation}
>     when $\epsilon  > |\mathcal{C}| \left\{ \left[\frac{1}{2\min_{1\leq c \leq |\mathcal{C}| }|\mathcal{S}_c^*|  } \log\left(\frac{|\mathcal{C}|}{\delta}\right) \right]^{1/2} +  \min_{1 \leq c \leq |\mathcal{C}|}\mathcal{M}_c^{(o)}(T)\right\}$, which completes the proof of (\ref{cls3_2}).
> ```
>
> > **Q8**: Could the authors test the model under different candidate set sizes, noise levels, or open-set cases (where true labels are missing)?
>
> **A8**: Thanks for your comment. As noted in Question 4, the current paper already varies q across multiple values per dataset (Table 1), which changes both noise level and expected candidate set size, and explicitly studies candidate set size in the competitive-noise experiment on Letter-High (Fig. 2(b)) by varying the sampling ratio from Top–K predictions. For open-set partial-label cases, we agree with the reviewer that this is an important direction. In the revision we will clarify that our benchmark setting follows the conventional closed-set PLL assumption and run additional experiments towards dropping the true label from the candidate set for a certain fraction of training samples.
>
> Thanks again for appreciating our work and for your constructive suggestions. Please let us know if you have further questions.

---

### Official Review · Reviewer_Nemd · 2025-10-30

**Soundness:** 4
**Presentation:** 3
**Contribution:** 3
**Rating:** 8
**Confidence:** 3

**Summary:**

This paper proposes a novel graph classification method in partial-label setting, which captures relations between graphs from both spatial and spectral views.

**Strengths:**

- The proposed method designs a novel way to compute spatial and spectral relations between graphs.
- The method's training complexity is linear in the number of edges and consistent with standard GNN-based methods.
- The method applies a binary mask to enable partial supervision.

**Weaknesses:**

- The equations lack explanations. Why do you design the spatial relations by Eq.(5), which consists of two components. Why do you use Eq.(7)  to compute X^(p) . What is the deeper meaning behind these equations.
- How to decide the binary mask M, as it desides how many data are used for supervision. Does this method sensitive to M and the ratio of supervised data.

**Questions:**

- I am not quite understand Figure 3, please provide more explanations.
- For Eq.16 and Eq. 17, which one is the final loss function?
- Why using different q for different datasets?

---

> ### Author Response · Authors · 2025-12-03
> **Response to Reviewer Nemd (I)**
>
> We are truly grateful for the time you have taken to review our paper, your insightful comments and support. Your positive feedback is incredibly encouraging for us! In the following response, we would like to address your major concern and provide additional clarification.
>
> > **Q1**: Why do you design the spatial relations by Eq.(5), which consists of two components.
>
> **A1**: Thanks for your comment. In Eq. (5), the first term represents the similarity between graphs i and j. However, considering that the true label of each graph is uncertain, using only cos(r_i,r_j) is not robust enough. When both graphs i and j have multiple candidate labels, the first term may be biased by noisy labels. The second term is the similarity between the average feature of graphs i and j and the prototype. When c is a common candidate label of graphs i and j, this prototype is the average feature encoding of all graphs containing c in the candidate label set. This can, to some extent, robustly pull the entire term toward the true label c^*. For example, consider that the candidate label set of graph i is {0,1,2,5} and the candidate label set of graph j is {0,1,3,4,5}. If only cos(r_i,r_j) is used, s_{ij}^{spa} is easily biased by labels 1 and 5. However, if there is another graph k with a candidate label set of {0,3}, then the second term can be pulled toward label 0, making it easier for the whole term to reach the maximum value at the true label 0.
>
>
> > **Q2**: Why do you use Eq.(7) to compute X^(p) . What is the deeper meaning behind these equations.
>
> **A2**: Thanks for your comment. Intuitively, Eq. (7) constructs a separable graph signal for each spectral mode, where the eigenvector determines the spatial pattern over nodes and the harmonic encoding encodes frequency characteristics, enabling frequency-aware yet size-agnostic graph representations. In spectral graph theory, each eigenpair (λ_p,u_p) corresponds to a mode on the graph: u_p(v) tells us how much node v participates in this mode, while λ_p encodes the frequency of the mode (how smooth/oscillatory it is). Eq. (7) enforces a clean separation between spatial and frequency information. Each band p is a “graph signal” whose node-wise amplitude pattern is dictated by u_p, while its frequency semantics are encoded in ρ(λ_p).
>
>
> > **Q3**: How to decide the binary mask M, as it desides how many data are used for supervision. Does this method sensitive to M and the ratio of supervised data.
>
> **A3**: Thank you for your comment. Technically, M \in {0,1}^{N \times C} encodes which labels are valid candidates for each training graph based on the partial label sets {S_i}^N_{i=1}, More specifically, we define M_{ic} = 1 if and only if c \in S_i and 0 otherwise. The mask is fixed by the problem setting and does not change during training. The amount of ambiguity is controlled by the noise parameter q, which determines how many incorrect labels are injected into each S_i. Our method is not overly sensitive to the amount of ambiguity encoded in M and the ratio of supervised data, empirically, Table 1 shows that PRISM degrades much more slowly than baselines as q increases from small to large values, including the hardest q=0.5 setting on ENZYMES. If q is so large that many candidate sets cover almost the entire label space, M becomes less informative and PRISM gradually reduces to a semi-supervised setting. In this extreme regime any PLL method would struggle; our results indicate PRISM remains robust over a wide practical range of ambiguity levels.
>
> > **Q4**: I am not quite understand Figure 3, please provide more explanations.
>
> **A4**: Thanks for your comment. Figure 3 visualizes class-specific attention maps generated by our spatial encoder on the ENZYMES dataset. Each subgraph represents a different graph instance, and the color of each node indicates its attention weight relative to the corresponding class prototype. Warmer colors signify higher contribution to the class-specific substructure reasoning. Although the graph structures vary, nodes with high attention consistently correspond to meaningful substructures such as motifs, central connectors, or functional cores, while irrelevant or peripheral nodes receive lower attention. This consistent focus within each class highlights the model’s ability to identify semantically aligned substructures across diverse graphs. We will revise the figure caption and main text to explain this more clearly in the revision.
>
> > **Q5**: For Eq.16 and Eq. 17, which one is the final loss function?
>
> **A5**: Thanks for your comment. Eq. (16) defines the candidate-constrained negative log-likelihood for a single view (e.g., spatial). Eq. (17) then combines the spatial and spectral terms. The final loss we optimize is Eq. (17), i.e., the sum of the spatial-view and spectral-view supervised losses \mathcal{L}_{final} = \mathcal{L}_{sup}^{spa} + \mathcal{L}_{sup}^{spe}.

---

> ### Author Response · Authors · 2025-12-03
> **Response to Reviewer Nemd (II)**
>
> > **Q6**: Why using different q for different datasets?
>
> **A6**: Thanks for your comment. We choose different numerical values of q so that the effective ambiguity level (expected number / fraction of candidate labels) is comparable across datasets with different numbers of classes. For example, if we used the same q for all datasets, then on a small-label dataset like ENZYMES(C=6), q=0.1 gives expected candidate set size ≈ 1.5 which is mild ambiguity, while on a large-label dataset like COIL-DEL(much larger C) the same q would produce very large candidate sets, effectively making the problem much noisier than on ENZYMES.
>
>
> Thanks again for appreciating our work and for your constructive suggestions. Please let us know if you have further questions.

---

### Official Review · Reviewer_kgnv · 2025-10-31

**Soundness:** 3
**Presentation:** 2
**Contribution:** 3
**Rating:** 4
**Confidence:** 4

**Summary:**

The paper aims to solve a novel graph classification problem known as partial label graph learning where labels of each graph are ambiguous with a set of labels provided but not precise label. The paper proposed a novel framework that aims to establish both graph-graph relation from spatial view and spectral view and denoise the label set from the established graph pair relations. For spatial view, the author proposed to align subgraph structure with the clustered class-centered substructure prototype and for spectral view, the author aims to build band-frequency based alignment. Empirical results suggest a clear improvement of proposed method compared with various baselines from graph neural network based models and partial label learning from computer vision fields. Ablation results suggest each component's effectiveness. The author also show theoretical analysis on the convergence of the label confidence matrix and training loss.

**Strengths:**

1.The paper discusses a novel problem called partial label graph learning problem which would be practical in real application setting where labels are ambiguous for actual datasets.
2. A novel framework is proposed that absorbs both spatial and spectral information for each graph to effectively capture useful information for denoise the noisy soft labels and train the classifier.
3. The theoretical analysis looks sound and comprehensive.
4. Empirical results suggest a clear improvements for proposed method compared with baselines from Graph neural network side and partial label learning benchmarks.

**Weaknesses:**

1. I think when it involves with the eigenvalue computation, especially for the dense and large-scale graph, the computation cost becomes expensive and prohibitive. This makes the spectral part of the framework not scalable. In the computation efficiency analysis, the author failed to discuss this important question and in my opinion falsely conclude the computation efficiency is comparable to standard GNN. For a dense and large-scale graph, the computation of eigenvalue and eigenvectors could be approaching O(n^2).
2. I find some parts of the paper lacks explanation of the symbols. For example, in the spectrum part, k seems to be an index, yet, in the computation analysis, k becomes the number of eigenvalues. M is referred to as the binary mask, but there is no definition on how it should be computed, which makes the paper hard to follow at some points.

**Questions:**

Please see weakness. I would like to know how the author considers their runtime on spectrum part and also how M is computed.

---

> ### Author Response · Authors · 2025-12-03
> **Response to Reviewer kgnv**
>
> We are truly grateful for the time you have taken to review our paper and your insightful review. Here we address your comments in the following.
>
> > **Q1**: I think when it involves with the eigenvalue computation, especially for the dense and large-scale graph, the computation cost becomes expensive and prohibitive. This makes the spectral part of the framework not scalable. In the computation efficiency analysis, the author failed to discuss this important question and in my opinion falsely conclude the computation efficiency is comparable to standard GNN. For a dense and large-scale graph, the computation of eigenvalue and eigenvectors could be approaching O(n^2).
>
>
> **A1**: Thank the reviewer for raising the issue of scalability. Our spectral encoder is designed for the graph classification regime, where we have many small to medium–sized graphs, not a single monolithic web-scale graph. From a theoretical perspective, by Eq. (6) we can see the eigenvalue computation is truncated and applied to sparse Laplacians, which makes the overhead modest compared to standard GNN training. From an experimental perspective, reviewer can see the following running time comparison chart of different methods.
>
>
> |Method|CIFAR10||
> |-|-|-|
> ||Time (min)| Acc (%)|
> |EdgePool|276.19|50.17|
> |Graph Transplant|169.35|53.79|
> |PRISM|155.51|58.29|
>
>
>
> > **Q2**: I find some parts of the paper lacks explanation of the symbols. For example, in the spectrum part, k seems to be an index, yet, in the computation analysis, k becomes the number of eigenvalues. M is referred to as the binary mask, but there is no definition on how it should be computed, which makes the paper hard to follow at some points.
>
> **A2**: Thank the reviewer for pointing out the ambiguity in the use of the symbol k. In the spectral module (Eq. (6)), k is intended to be a local dummy index inside the harmonic expansion, while the number of harmonic components (equivalently, frequency bands) is denoted by T. In the current draft, the compact notation cos[sin(kλ),cos(kλ)]^T_{k=1} may give the impression that k is a global hyperparameter, which is not our intention. We will revise this part to remove the overloaded use of k and to explicitly state that T denotes the number of frequency bands. In the complexity analysis of the spectral branch, the notation “top-k eigenvalues” is meant to refer to the number of eigenpairs retained from the Laplacian, which should be denoted by a separate symbol, in the revised manuscript, we will use "T" consistently as the number of spectral bands in the harmonic expansion and use a different symbol (e.g. P) for the number of Laplacian eigenpairs retained, when we discuss computational cost.
> As for the definition and construction of the binary mask M, the reviewer is absolutely right that, as written, the binary mask M is only mentioned but not explicitly defined. Technically, M \in {0,1}^{N \times C} encodes which labels are valid candidates for each training graph, based on the partial label sets {S_i}^N_{i=1}, More precisely, we define M_{ic} = 1 if and only if c \in S_i and 0 otherwise. The mask is fixed by the problem setting and does not change during training. We appreciate the comment on readability. In revising the paper, we will explicitly define M and explain in one sentence that it is derived from the candidate label sets.
>
> In light of these responses, we hope we have addressed your concerns, and hope you will consider raising your score. If there are any additional notable points of concern that we have not yet addressed, please do not hesitate to share them, and we will promptly attend to those points.

---

### Author Response · Authors · 2025-12-04
**Summary of Rebuttal**

Dear ACs, SACs, PCs, and Reviewers,

We sincerely appreciate the great efforts and the constructive suggestions you have provided once again! Your feedback has been invaluable in helping us improve the clarity and quality of the paper.

The reviewers recognize that the problem addressed by our work is **important and practically relevant** (Reviewer kgnv, Reviewer Qmcu, Reviewer Nyo7). They note that our framework is **conceptually coherent and well-motivated** (Reviewer Qmcu), that the integration of spatial and spectral relational modeling is **novel** (Reviewer kgnv, Reviewer Nemd), and that the theoretical analysis is **sound and comprehensive** (Reviewer kgnv). The empirical results are viewed as **strong** (Reviewer kgnv, Reviewer Qmcu, Reviewer Nyo7), with clear improvements over competitive baselines, and the method is regarded as **modular and easy to adapt** (Reviewer Nyo7).

During the rebuttal, we addressed all concerns raised by the reviewers. We provided detailed clarifications on the scalability of the spectral encoder, the interpretation of key equations, the construction of the binary mask and filtering mechanism, and the complementary roles of the spatial and spectral views. We also introduced a refined convergence theorem that avoids strong assumptions and better reflects the learning dynamics in practice. We hope these clarifications have helped present a more complete picture of the proposed framework. Thank you again for your thoughtful feedback and support.

Best regards,

The Authors

---

### Meta-Review · Area_Chair_GaGk · 2025-12-10

**Summary:**

This paper introduces PRISM, a framework that resolves label ambiguity in partial-label graph learning by combining spatial substructure cues, spectral multi-band features, and relational label propagation. Reviewers found the problem meaningful and the empirical results strong, though they initially questioned the scalability of the spectral module, the clarity of key definitions, and the strength of the theoretical assumptions. The authors addressed these issues by clarifying notation, defining the mask and confidence filtering mechanism, refining the complexity analysis, and providing a more realistic convergence theorem, which alleviated most concerns. Overall, I recommend acceptance, while suggesting that the final version further illustrate how the two views complement each other and more clearly delimit the method’s scalability.

**Reviewer Concerns:**

The rebuttal effectively resolves several concerns shared across reviewers, including the unclear definition of the binary mask and confidence filtering, the overloading of notation in the spectral module, and the lack of a precise complexity breakdown. The clarification that alignment is done at the graph-level rather than through subgraph matching also addresses questions about spatial alignment cost. The refined convergence theorem responds to worries about the strength of earlier assumptions. What remains less fully resolved are concerns about scalability on very large or dense graphs and the need for more concrete evidence showing how the spatial and spectral views correct each other in practice.

**Reviewer Scores:**

Reviewer kgnv began with a score of 4 and stated they would not mind if the paper were accepted; given the clarifications on complexity and notation, their score would likely move up slightly to a weak accept. Reviewer Nemd started at 8 and raised mainly explanatory questions; these were addressed directly, so their score would probably remain at 8. Reviewer Qmcu began at 6 and expressed that acceptance or rejection would both be acceptable; the rebuttal answered their core concerns about alignment and theory, so their score would likely rise to a clearer accept. Reviewer Nyo7 began at 4 and had reservations about theory, complementarity, and complexity; the authors’ responses reduce but do not fully remove these doubts, so the score might increase modestly to a borderline accept.

---

### Decision · Program_Chairs · 2026-01-26

Accept (Poster)